# Unfolding the Black Box of Recurrent Neural Networks for Path Integration

Tianhao Chu[1,†], Yuling Wu[2,†], Neil Burgess[3,4], Zilong Ji[1,3], Si Wu[1,2,5,*]

[1]School of Psychological and Cognitive Sciences, Key Laboratory of Machine Perception (Ministry of Education), Peking University, China.
[2] Peking-Tsinghua Center for Life Sciences, Academy for Advanced Interdisciplinary Studies, Peking University, China.
[3]Institute of Cognitive Neuroscience, Department of Neuroscience, Physiology and Pharmacology, University College London, UK
[4]UCL Queen Square Institute of Neurology, University College London, UK
[5]Beijing Key Laboratory of Behavior and Mental Health, IDG/McGovern Institute for Brain Research, Center of Quantitative Biology, Peking University, China.
[†]These authors contributed equally to this work.
[*]Correspondence: siwu@pku.edu.cn

## Abstract

Path integration is essential for spatial navigation. Experimental studies have identified neural correlates for path integration, but exactly how the neural system accomplishes this computation remains unresolved. Here, we adopt recurrent neural networks (RNNs) trained to perform a path integration task to explore this issue. After training, we borrow neuroscience prior knowledge and methods to unfold the black box of the trained model, including: clarifying neuron types based on their receptive fields, dissecting information flows between neuron groups by pruning their connections, and analyzing internal dynamics of neuron groups using the attractor framework. Intriguingly, we uncover a hierarchical information processing pathway embedded in the RNN model, along which velocity information of an agent is first forwarded to band cells, band and grid cells then coordinate to carry out path integration, and finally grid cells output the agent location. Inspired by the RNN-based study, we construct a neural circuit model, in which band cells form one-dimensional (1D) continuous attractor neural networks (CANNs) and serve as upstream neurons to support downstream grid cells to carry out path integration in the 2D space. Our study challenges the conventional view of considering grid cells as the principal velocity integrator, and supports a neural circuit model with the hierarchy of band and grid cells.

## 1 Introduction

Path integration refers to the process of computing the position of an agent by continuously integrating self-motion information of the agent over time [1]. It is fundamental for spatial navigation in both humans and rodents, and also plays an important role in robot control [2].

Experimental studies have shown that grid cells in the medial entorhinal cortex (MEC) are deeply involved in path integration [3]. These cells exhibit periodic hexagonal firing patterns in the two-dimensional space, providing a metric representation for implementing path integration [1, 4]. Disruption of grid cells through medial septal inactivation or entorhinal lesions impairs spatial memory and navigation based on path integration [3, 5]. Despite these experimental evidence,

exactly how the neural system performs path integration remains largely unknown, and there are unresolved issues. First, in addition to grid cells, experimental data has shown that there exists a rich diversity of spatially modulated cell types in the MEC [6, 7, 8, 9]. In particular, band cells, found in parasubiculum (an upstream region of MEC), exhibit periodic firing patterns that can be modeled as a superposition of bands, and they are hypothesized to perform path integration in a 1D space [9, 10]. However, the functional relationship between grid and band cells remains to be clarified. Second, neural circuit models have been proposed to implement path integration. The conventional models consider only grid cells, and in order to implement 2D path integration, these models either consider a large amount of direction-conjunctive grid cells [11, 12, 13] or assume that the recurrent connections between grid cells are tuned instantly by the velocity signal [14]. A different model considering the hierarchy of band and grid cells for implementing path integration was proposed [15, 16]. The question of which model is biologically more plausible remains debated.

**Related RNN-based navigated studies**. Recently, AI approaches have been used to study the association between path integration and grid cells. Specifically, recurrent neural networks (RNNs) were trained to perform spatial navigation to ascertain the involvement of grid cells in path integration, and inconclusive results have been reported so far. In one line of works, it was found that training an RNN based solely on motion velocity in a supervised way (i.e., reconstructing the place cell activity at the given location from pure velocity input) can lead to the emergence of grid-cell-like activity pattern in the RNN, especially under the biological constraints such as non-negative firing rate [17, 18]. This result implies that grid cells are associated with path integration. Building on this work, subsequent studies have extended the supervised training paradigm to more unsupervised-guided ones, using objective functions derived from group theory, information efficiency, and actionability, emphasizing the isometric structure, high coding capacity, and integrative functionality of grid cells [19, 14, 20]. These studies have systematically explored the conditions under which grid-like representations emerge in RNNs, examining how the network structure, the training objective, and biological priors jointly shape grid-like representations. In the other line of works, the necessity of grid cells for path integration is challenged. For instance, [21] demonstrated that grid-like firing patterns can arise as the principal components of place cell activities under the non-negativity constraint, without path integration involved. More recently, [22] argued that grid cells may emerge from a principle of efficient pattern formation for reconstructing place fields, rather than from the demand of path integration. Supporting this view, ablation studies of trained RNNs have shown that removing velocity inputs to high grid-score units has negligible impact on the path integration performance [23, 24]. Overall, current RNN-based studies have primarily focused on exploring the conditions under which grid cells emerge and on the necessity of grid cells for path integration. However, they have not addressed the question of how a trained RNN implements path integration, nor examined the functional role of band cells in this process [18, 25]. The internal mechanisms of the trained RNN model thus remain a black box.

In this work, we take the same approach of training RNNs to perform path integration, but our goal is different, that is, we aim to unveil the detailed mechanism concealed in the trained RNNs, and from which, to ascertain the neural mechanism of path integration. To achieve this goal, we will borrow neuroscience prior knowledge and methods to unfold the black box of the trained RNNs in much more detail than in previous studies. These include: clarifying neuron types based on their receptive fields, dissecting information flows between neuron groups by pruning their connections, and analyzing internal dynamics of neuron groups in the attractor framework. It turns out that there exists a hierarchical information processing pathway embedded in the RNN model, along which velocity information is first forwarded to band cells, band and grid cells then coordinate to carry out path integration, and finally grid cells output the location of the agent. Inspired by the RNN-based study and previous mechanistic models [16, 26], we construct a hierarchical neural circuit model for path integration in the brain, in which band and grid cells form 1D and 2D continuous attractor neural networks (CANNs), respectively, and they coordinate to accomplish path integration. Contemporary models often assume that grid cells are primarily responsible for path integration [11, 12, 13]. Our study challenges this view and suggests that the hierarchy of band and grid cells is necessary. Our model also makes predictions testable in neuroscience experiments.

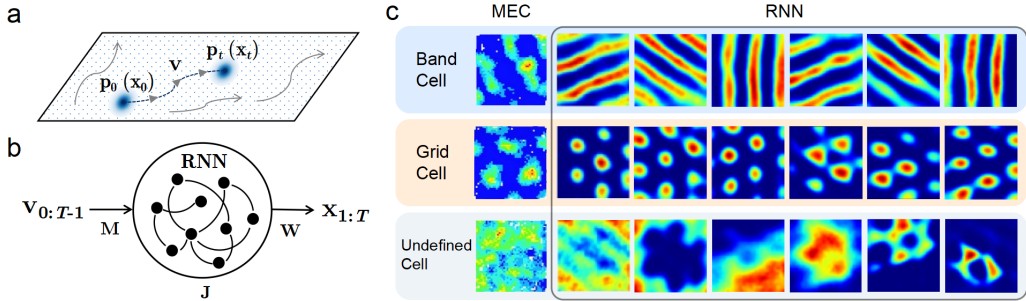

Figure 1: (a) Illustration of the path integration task. (b) The structure of the RNN model. (c) Examples of band cell, grid cell and undefined cell found in MEC (middle panel, adapted from [9]) and in the trained RNN (right panel).

## 2 Neuroscience for AI: unfolding the black box

### 2.1 An RNN model for path integration

We adopt a well-established paradigm to train RNNs for path integration [17, 22], obtaining the key results as reported here, which are further validated by consistent results from other modeling frameworks (see Supplement Information (SI) E). As illustrated in Fig. 1a, an agent is moving in a square environment with velocity $\mathbf{v}_t \in \mathbb{R}^2$. Mimicking place cells' position encoding in the neural system, the agent's location $\mathbf{x}_t$ is represented by a bump vector $\mathbf{p}_t \in \mathbb{R}^{N_p}$ in the 2D space centered at $\mathbf{x}_t$, with $N_p$ the number of place cells. The task of path integration is formulated as: given the initial location $\mathbf{x}_0$ of the agent, the RNN model infers its subsequent positions $\mathbf{x}_{1:T}$ by using only the self-motion information of the agent, i.e., the sequence of velocity vectors $\mathbf{v}_{0:T-1}$; no other position cue is available. Within the conceptual framework of the MEC-HPC loop, the model's information flow can be interpreted as follows: the initial state of the RNN (representing a nascent MEC population) is set by a projection from the place cell population (representing hippocampus, HPC), i.e., $\mathbf{r}_0 = \mathbf{M}_0\mathbf{p}_0$, where $\mathbf{p}_0$ corresponds to the initial location $\mathbf{x}_0$ and the matrix $\mathbf{M}_0 \in \mathbb{R}^{N_R \times N_P}$, with $N_R$ the number of neurons in the RNN (Fig. 1b). This initialization embodies a critical **HPC $\rightarrow$ MEC projection** signal that anchors the path-integration process to an environmental cue.

At each time step, the RNN receives the agent's instantaneous velocity signal $\mathbf{v}_t$ and updates its internal state following the recurrent dynamics, which is written as,

$$\mathbf{r}_{t+1} = \sigma\left(\mathbf{J}\mathbf{r}_t + \mathbf{M}\mathbf{v}_t\right), \tag{1}$$

where $\mathbf{J} \in \mathbb{R}^{N_R \times N_R}$ is the recurrent connection matrix between neurons in the RNN, the matrix $\mathbf{M} \in \mathbb{R}^{N_R \times 2}$, and $\sigma(\cdot)$ is the ReLU function, enforcing non-negative firing rates of neurons.

At each time step, the agent's location is read out by,

$$\hat{\mathbf{p}}_t = \mathbf{W}\mathbf{r}_t. \tag{2}$$

where the read-out matrix $\mathbf{W} \in \mathbb{R}^{N_p \times N_R}$.

The free parameters of a RNN, including $\mathbf{J}$, $\mathbf{M}$, $\mathbf{W}$, and $\mathbf{M}_0$, are optimized using BPTT through minimizing the cross-entropy loss between the network outputs $\hat{\mathbf{p}}_t$ and the ground truth $\mathbf{p}_t$ for all time steps $t = 1, \ldots, T$ and for all randomly generated motion trajectories. The details of the model training and fixed parameters are given in SI A. The trained RNN model effectively implements path integration, with quantitative results (SI D) indicating that it achieved competent performance on the task.

### 2.2 Interpreting the trained RNN model with neuroscience

It is straightforward to train the RNN model well to achieve a good path integration performance, but unveiling the underlying computational mechanism is challenging, which is known as the black box problem of AI-trained models. Without prior knowledge, we have no clue to interpret the data. Here, we borrow neuroscience prior knowledge and methods to unfold the black box.

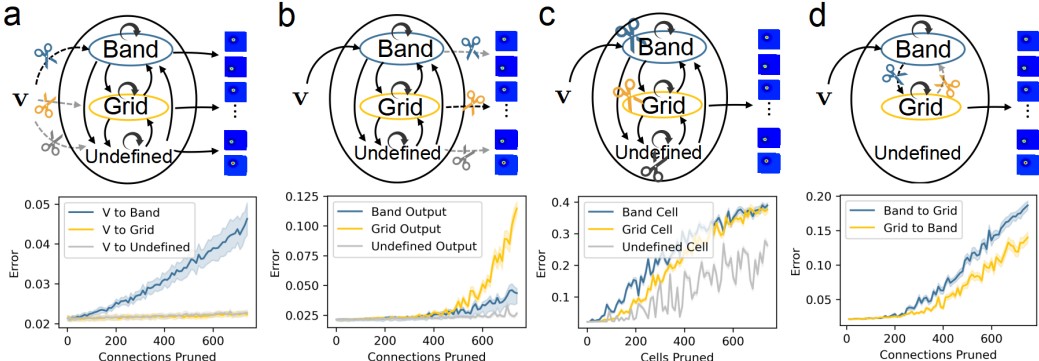

Figure 2: Results of pruning the RNN model. Four pruning operations are carried out step-by-step, with each step building upon the previous one. Upper panels: (a) Pruning velocity inputs to each cell group one-by-one; (b) Pruning read-out connections from each cell group one-by-one; (c) Pruning each cell type one-by-one; (d) Pruning the connections from band to grid and the connections from grid to band cells, respectively. Lower panels: Path integration error vs. the number of connections/cells pruned in each step. The path integration error is measured by the standard division of location prediction errors divided by the arena size. For calculation details, see SI C, E.

### 2.2.1 Clarifying neuron types

Our first step is to clarify neuron types in the trained RNN based on their receptive fields. Following the common practice in neuroscience, we first compute the spatial tuning maps of neurons in the trained RNN. Specifically, we consider that the agent performs random walks in the environment, covering the whole arena, and the corresponding velocity signals are used as inputs to stimulate neuronal responses in the RNN. We record the activities of each neuron at all locations, which gives a 2D heatmap for each neuron, reflecting their spatial tuning characteristics.

Inspired by the spatial tuning properties of grid and band cells observed in the neural system, we search for analogous unit types in the trained RNN. To classify neuron types quantitatively, we adopt established criteria from neuroscience (see SI B). We have identified three mutually exclusive types based on neuronal receptive fields (Fig. 1c, right panel), which are: 1) Grid cell, which exhibits the hexagonally patterned firing field; 2) Band cell, whose receptive field consists of multiple equally spaced parallel lines (the so-called bands) (see more details below); and 3) Undefined cell, which displays neither grid-like nor band-like spatial tuning. All three cell types emerge robustly in the trained RNNs (see SI E). For example, in one trained RNN with $N_R = 4096$ units, we identify 764 grid cells, 764 band cells, and 2568 undefined cells).

### 2.2.2 Dissecting information flows between neuron groups

After identifying neuron groups, we further investigate how information is propagated between them by applying pruning studies. We carry out four step-by-step pruning operations (see SI C), with each of them building upon the previous one and aiming to probe one aspect of path integration. See SI E for reproducibility results.

**Step 1: Pruning velocity inputs**. To identify which cell types are primarily responsible for receiving and processing velocity inputs, we selectively prune the afferent velocity inputs to each cell group one-by-one (Fig. 2a, upper panel). We find that pruning velocity inputs to band cells leads to a significant increase in the path integration error, while pruning velocity inputs to other groups has little effect (Fig. 2a, lower panel). This indicates that the velocity information is predominantly processed by band cells, and we remove velocity inputs to other cell types in the followed pruning.

**Step 2: Pruning read-out connections**. To identify which cell types are primarily responsible for outputting the path integration result, we selectively prune the read-out connections from each cell group one-by-one (Fig. 2b, upper panel), and find that pruning the read-out connections from grid cells have the largest effect on increasing the path integration error (Fig. 2b, lower panel). This

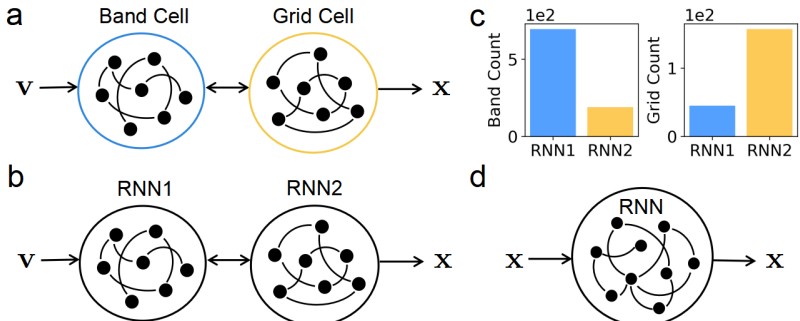

Figure 3: (a) The hierarchical path integration pathway embedded in trained RNNs. (b) A new RNN model taking the hierarchical pathway as a prior. (c) Neuron types found in the newly trained RNN. (d) An RNN model for a location reconstruction task. For simulation details, see SI C.

indicates that grid cells are predominately responsible for outputting the path integration result, and we remove the read-out connections from other cell types in the followed pruning.

**Step 3: Pruning cell types.** To evaluate the relative contributions of different cell types on path integration, we compare the model performances after pruning one of them in equal numbers (Fig. 2c, upper panel). We find that pruning band or grid cells causes a substantial drop in the model performance, whereas pruning undefined cells has a much less effect (Fig. 2c, lower panel). This suggests that band and grid cells play much more important roles than undefined cells in path integration, and we remove them in the followed pruning.

**Step 4: Pruning connections between band and grid cells.** To probe the functional relationship between band and grid cells in path integration, we separately prune the connections from band to grid cells and the connections from grid to band cells (Fig. 2d, upper panel). We find that pruning either type of connections has a big impact on the model performance (Fig. 2d, lower panel), indicating that the bidirectional, cooperative interactions between two cell types are critical for path integration.

## 2.3 A hierarchical path integration pathway embedded in the RNN

Combining pruning results while discarding less relevant components, we identify a hierarchical path integration pathway embedded in the trained RNN (Fig. 3a), along which velocity information is first received and processed by band cells; band and grid cells then interact via reciprocal connections to integrate movement information; finally grid cells output the agent location in the 2D space.

It is surprising that such a well-organized pathway naturally emerges in training a RNN for a path integration task. To further validate that this hierarchical pathway constitutes the functional core structure, we conduct two additional experiments. In one experiment, we construct a new RNN model which takes the hierarchical pathway as a prior (Fig. 3b), that is, the first module receives velocity inputs, intending to play the role of band cells; the second module outputs the agent location, intending to play the role of grid cells; the two modules are reciprocally connected, intending to mimic the interactions between band and grid cells. We train this new model with the same path integration task and have two key observations as expected, which are: 1) the new model achieves a performance comparable to the original model; 2) band and grid cells emerge naturally as the dominant neuron types in the first and second modules, respectively (Fig. 3c). In the other experiment, we change the task from path integration to pure location reconstruction (Fig. 3d), i.e., the RNN receives the agent's location inputs and reconstruct them; no velocity information is available. After training the RNN with the new task, we observe only grid cells without band cells (see SI C), indicating the necessity of path integration for the emergence of band cells. Together, these two additional experiments further reinforce our finding that the hierarchical pathway, characterized by the hierarchy of band and grid cells, and their reciprocal interactions, is the core structure for path integration.

### 2.3.1 Continuous attractor dynamics of band cells

We now delve deeper into the internal structure and dynamics of band cells in the trained RNN.

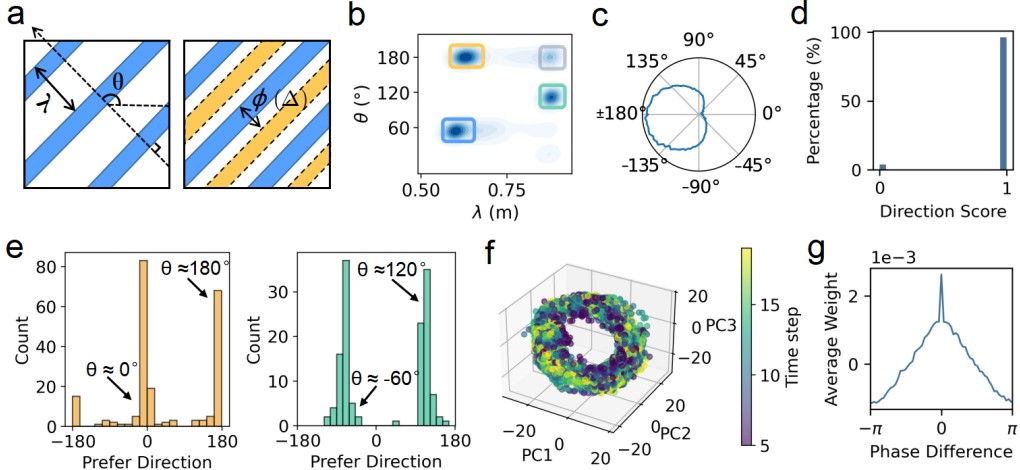

Figure 4: Properties of band cells in the trained RNN. (a) Illustration of the receptive field of an idealized band cells. Left panel: the receptive field consists of parallel bands, which are equally spaced with spacing $\lambda$ and have the same orientation $\theta$. Right panel: the positional separation between two band cells having the same spacing and orientation (blue and yellow) is quantified by a phase $\phi$, whose value is given by $\phi = (\Delta/\lambda) \times 2\pi$, with $\Delta$ the offset between the adjacent bands of two band cells. (b) Band cells in the trained RNN are grouped into four clusters according to their spacing $\lambda$ and orientation $\theta$. (c) An example band cell with direction-tuning at $\pi$. (d) Distribution of directional scores (a measurement of the direction-tuning level) of band cells in an example cluster. (e) Histograms of preferred directions of band cells in two example clusters, which have a bimodal shape peaked at either the cluster orientation $\theta$ or its opposite direction $\theta + \pi$. (f) A 3D isomap embedding of the population activities of band cells in an example group with the same spacing and direction tuning. Coloured by time steps 5–20 of 200 trajectories. (g) The averaged connection profile of band cells as a function of their phase difference. For detail calculations, see SI B.

Fig. 4a displays the receptive field of an idealized band cell, which consists of multiple equally spaced parallel lines (bands), on which the cell's response is invariant. A band cell can be characterized by three parameters: 1) spacing $\lambda$, which is the distance between adjacent bands; 2) orientation $\theta$, which is the orientation of bands in the 2D space; 3) phase $\phi$, which defines the positional separation between band cells having the same spacing and orientation. Denote $\Delta \in (-\lambda/2, \lambda/2)$ the offset between the adjacent bands of two band cells, their phase difference is given by $\phi = (\Delta/\lambda) \times \pi \in (-\pi, \pi)$.

First, we inspect the clustering property of band cells in the trained RNN. Using Fourier analysis, we extract the dominant spatial frequency of the receptive field of each band cell and obtain the cell's spacing and orientation (see SI B for details). We find that band cells can be grouped into four well-separated clusters based on their spacing and orientation (Fig. 4b). For each cluster, we further differentiate cells' direction-tuning by calculating how the averaged response of a band cell varies with motion direction of the agent. Interestingly, we find that in each cluster, the majority of band cells exhibit clear direction-tuning (Fig. 4c-d), and they constitute two sub-groups with preferred directions along either the cluster orientation $\theta$ or its opposite $\theta + \pi$ (Fig. 4e).

Second, inspired by recent successes in modeling spatially tuned neurons in the hippocampal-entorhinal system using Continuous Attractor Neural Networks (CANNs) [27, 28, 29, 11], we set out to investigate whether the low-dimensional manifold of neural activity and its underlying connectivity conform to CANN principles. Second, to uncover the latent structure and dynamics of band cells in each group having the same spacing and direction-tuni ng, we performed dimensionality reduction dimensionality reduction analysis (Isomap, details shown in SI B) on the activity trajectories of the group during path integration. Strikingly, we observed that the low-dimensional embedding of neural trajectories in each group forms a continuous ring (Fig. 4f), consistent with the dynamics of a 1D CANN with periodicity. We also inspected the recurrent connectivity pattern between band cells, looking for the signature of the CANN structure. For each band cell group having the same spacing and direction-tuning, we calculated the averaged connection weights between cell pairs

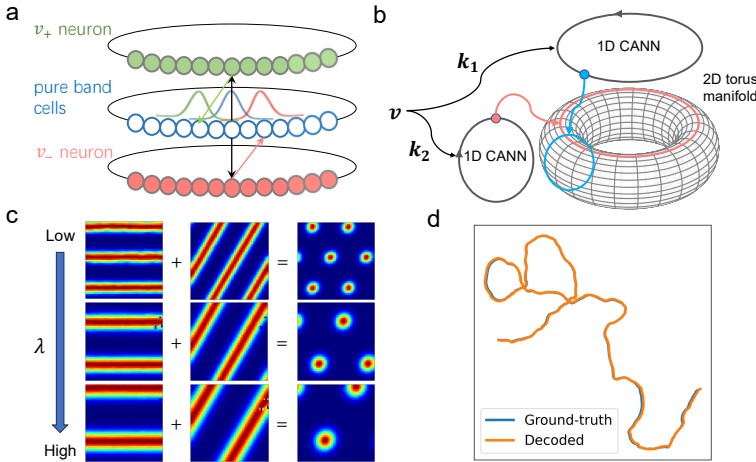

Figure 5: A neural circuit model for path integration. (a) The structure of a band cell module, which consists of three sub-populations: pure band cells, direction-conjunctive band cells $v^+$, and direction-conjunctive band cells $v^-$. A band cell module performs 1D path integration in the form of continuous attractor dynamics, along either the module orientation $v^+ = \theta$ or the opposite direction $v^- = \theta + \pi$. (b) Two band cells modules with different orientations support the grid cell module of the 2D CANN structure to accomplish path integration in the 2D space. (c) Examples of band and grid cells in the neural circuit model. (d) Inferred trajectory of the neural circuit model closely matches the ground-truth trajectory, demonstrating the path integration capability of the model.

as a function of their phase differences. The result shows that the connectivity profile exhibits an approximately symmetric shape with weight decreasing with phase difference (Fig. 4g), reminiscent of the center-surround connection pattern in a 1D CANN [30].

In summary, the above study reveals that band cells in the trained RNN constitute modules of different spacing and direction-tuning, and each module displays the 1D CANN-like structure and dynamics.

## 3 AI for neuroscience: the neural mechanism for path integration

The above RNN-based navigation study also gives us insight into the path integration mechanism in the neural system. Band cells were reported in a neuroscience experiment [9], and an early computational model [15, 16] (in which band cells were called Velocity-Controled-Oscillator (VCO)) was also proposed, which considered that band cells are responsible for 1D path integration, and multiple modules of band cells of different orientations jointly support grid cells to execute path integration in the 2D space. Our RNN-based study tends to support this view, and challenges the conventional models of relying only on grid cells for path integration.

### 3.1 A neural circuit model for path integration

Inspired by the findings in the trained RNN, we construct a neural circuit model for path integration. It has the similar idea as that in [15, 16], but have different mathematical formulations. In the below, we introduce the main structure of the model, with details presented in SI F.

#### 3.1.1 The modules of band cells

Motivated by the properties of band cells found in the trained RNN and the 1D head-direction system in Drosophila [31, 32], we propose a theoretical model for band cells. Specifically, we consider that band cells are clustered into multiple modules with varying spacing $\lambda$ and orientation $\theta$. Each module consists of three sub-populations (Fig. 5a), which are: 1) pure band cells, 2) direction-conjunctive band cells $v^+$, and 3) direction-conjunctive band cells $v^-$, and they are aligned on a 1D manifold according to their preferred phases $\phi \in (-\pi, \pi]$. Each module of band cells implements path

integration in a 1D direction, along either the module orientation $v^+ = \theta$ or the opposite direction $v^- = \theta + \pi$.

The pure band cell population forms a 1D CANN, whose dynamics is given by,

$$\tau\frac{\partial u_b(\phi,t)}{\partial t} = -u_b(\phi,t) + \rho\int_{-\pi}^{\pi} J_b(\phi,\phi')\, r_b(\phi',t)\, d\phi' + \rho\sum_{m=\pm}\int_{-\pi}^{\pi} W_m(\phi,\phi')\, r_m(\phi',t)\, d\phi' + I_g(\phi,t),$$
(3)

where $u_b(\phi,t)$ and $r_b(\phi,t)$ denote the synaptic input and firing rate of pure band cells at $\phi$, respectively. $J_b(\phi,\phi')$ denotes the recurrent connections between band cells, and $W_m$, with $m = \pm$, denotes the connections from direction-conjunctive cells $v^m$ to pure band cells. $I_g$ denotes the input from grid cells (to be defined below). The firing rate of pure band cells is given by $r_b(\phi,t) = u_b^2(\phi,t)/\left[1 + k_b\rho\int_{-\pi}^{\pi} u_b^2(\phi',t)\, d\phi'\right]$, with $k_b$ regulating the global inhibitory strength to ensure a stable bump activity of band cells.

Conjunctive band cells $v^\pm$ receive excitatory inputs from pure band cells and are tuned by the moving direction of the agent, whose dynamics are written as,

$$\tau\frac{\partial u_\pm(\phi,t)}{\partial t} = -u_\pm(\phi,t) + w_b \cdot r_b(\phi,t), \tag{4}$$
$$r_\pm(\phi,t) = [g_0 + (\mathbf{v}(t)\cdot\mathbf{k})u_\pm(\phi,t)]_+, \tag{5}$$

where $u_\pm(\phi,t)$ and $r_\pm(\phi,t)$ denote the synaptic input and firing rate of conjunctive cells at phase $\phi$, respectively. $w_b$ denotes the connection from pure band to conjunctive cells at the same phase. $\mathbf{k} = (\sin\theta/\lambda, \cos\theta/\lambda)$ denotes a module-specific vector, and the projection of the agent velocity $\mathbf{v}(t)$ on it, $\mathbf{v}(t)\cdot\mathbf{k}$, defines the moving speed of the agent along the orientation $\theta$. $g_0$ is a baseline constant. $[\cdot]_+$ denotes rectification. Eq.5 reflects that the firing rates of conjunctive cells contain the projected 1D speed information of the agent along the module orientation [31].

Importantly, we set the recurrent connections between pure band cells to be symmetric and translation-invariant, i.e., $J_b(\phi,\phi') = J_b^0/\left(\sqrt{2\pi}\sigma_b\right)\exp\left[-|\phi-\phi'|_\pi^2/(2\sigma_b^2)\right]$, while the connections from conjunctive cells to pure band cells are offset with a constant value $\delta$, i.e., $W_\pm(\phi,\phi') = W_b^0/(\sqrt{2\pi}\sigma_b)\exp\left[-|\phi-\phi'\mp\delta|_\pi^2/(2\sigma_b^2)\right]$. Together, three populations of band cells in a module achieve 1D path integration along the module orientation [31].

### 3.1.2 The module of grid cells

A single band cell module supports path integration in two directions $\theta$ and $\theta + \pi$. To realize 2D path integration, grid cells integrate the outputs of band cell modules of different orientations. With loss of generality, we consider only two band cell modules in the present study (Fig. 5b). The orientations of two modules are denoted as $\theta_1$ and $\theta_2$, respectively, and they are $60°$ apart. Grid cells are aligned on a 2D toroidal manifold with a phase vector $\boldsymbol{\phi} = (\phi_1,\phi_1)$. They form a 2D CANN with dynamics given by,

$$\tau_g\frac{\partial u_g(\boldsymbol{\phi},t)}{\partial t} = -u_g(\boldsymbol{\phi},t) + \iint_{-\pi}^{\pi} J_g(\boldsymbol{\phi},\boldsymbol{\phi}')\, r_g(\boldsymbol{\phi}',t)\, d\boldsymbol{\phi}' + I_1(\phi_1,t) + I_2(\phi_2,t), \tag{6}$$

where $u_g(\boldsymbol{\phi},t)$ and $r_g(\boldsymbol{\phi},t)$ are the synaptic input and firing rate of grid cells at phase $\boldsymbol{\phi}$. The recurrent connections between grid cells are symmetric and translation-invariant, i.e., $J_g(\boldsymbol{\phi},\boldsymbol{\phi}') = J_0^g/(2\pi\sigma_g^2)\exp\left[-||\boldsymbol{\phi}-\boldsymbol{\phi}'||_g^2/(2\sigma_g^2)\right]$, with $||\cdot||_g$ denotes circular distance on the torus and $\sigma_g$ controlling the bump activity size. The firing rate of grid cells also follows divisive normalization, i.e., $r_g(\boldsymbol{\phi},t) = u_g^2(\boldsymbol{\phi},t)\left[1 + k_g\iint u_g^2(\boldsymbol{\phi}',t)\, d\boldsymbol{\phi}'\right]$. $I_1(\phi_1,t)$ and $I_2(\phi_2,t)$ represent the inputs from two band cell modules having orientations $\theta_1$ and $\theta_2$, respectively (to be defined below).

### 3.1.3 Interactions between band and grid cells

We set the reciprocal connections between band and grid cells to be symmetric, which are written as,

$$W_{gb}(\phi_k^b,\boldsymbol{\phi}^g) = \frac{W_{gb}^0}{\sqrt{2\pi}\sigma_{gb}}\exp\left[-\frac{|\phi_k^b-\phi_k^g|_\pi^2}{2\sigma_{gb}^2}\right], \quad k = 1,2, \tag{7}$$

where $k = 1, 2$ indexes the band cell module.

Thus, the inputs from grid cells to each band cell module are given by,

$$I_g(\phi_k^b, t) = \iint_{-\pi}^{\pi} W_{gb}(\phi_k^b, \phi^{g'}) \, r_g(\phi^{g'}, t) \, d\phi^{g'}, \quad k = 1, 2, \tag{8}$$

and the inputs from each band module to grid cells are given by

$$I_k(\phi_k, t) = \int_{-\pi}^{\pi} W_{gb}(\phi^g, \phi_k^{b'}) \, r_b(\phi_k^{b'}, t) \, d\phi_k^{b'} \quad k = 1, 2. \tag{9}$$

The forward connections from band to grid cells enable the grid cell module to integrate 1D motion trajectories into 2D, while the feedback connections from grid to band cells enable the circuit to amend errors. They jointly support accurate path integration in the 2D space.

## 3.2 Simulation results

We carry out simulations to test the performance of the neural circuit model (for details, see SI F). We observe that band and grid cells of varying spacings and orientations appear in the hand-designed model (Fig. 5c). The moving trajectory inferred by the model agrees well with ground truth, demonstrating the path integration capacity of the model (Fig. 5d).

It should be noted that the path integration capability demonstrated here builds upon well-established computational frameworks [11, 31], and thus we do not perform additional tests on its robustness in this regard. Furthermore, the absence of accumulated error in this simulation is due to the idealized setting where neither input signals nor the model contain any random noise. In more realistic scenarios where noise is present, small errors would accumulate over time, leading to gradual drift in the estimated position; corresponding results under noisy conditions are provided in SI F.

## 4 Discussion

In the present study, by using neuroscience prior knowledge and methods, we unfold the black box of RNNs trained to perform a path integration task, and we identify a hierarchical path integration pathway embedded in trained RNNs. Using a combination of analyses (neuron type classification, connection pruning, attractor dynamics inspection et al.), we uncover that along this hierarchical pathway, velocity information of an agent is first conveyed to band cells, band and grid cells then coordinate to carry out path integration, and finally grid cells output the agent location. Furthermore, we reveal that band cells form multiple modules of varying spacing and orientation, and each module constitutes a 1D CANN-like structure. Inspired by the RNN-based study and other neuroscience evidence, we formulate a neural circuit model for path integration. In this model, band cells form multiple functional modules in the form of 1D CANNs, and each of them is responsible for path integration along the module orientation; grid cells form a 2D CANN, which integrates the 1D results of band cell modules to carry out 2D path integration and meanwhile provides feedback to band cells to correct errors. We demonstrate that this computational model works well.

**Related neural circuit models.** Previous neural circuit models for path integration based only on grid cells require a large amount of direction-conjunctive grid cells, whereby these conjunctive cells receive velocity inputs and shift the activity of the downstream pure grid cells along the same direction to perform location updating [11]. Compared to this model, which requires $N^2$ pure grid cells and at least $4N^2$ conjunctive grid cells to represent four allocentric directions (north, south, east, and west), the hierarchical model we propose requires only $2N$ pure band cells, $4N$ conjunctive band cells ($2N$ per each module), and $N^2$ grid cells. Thus, the total number of neurons required to perform path integration is significantly reduced from $5N^2$ to $6N + N^2$, without sacrificing the performance. This computational efficiency offers new insight for the advantage of employing the combination of band and grid cells. To reduce the neuron number, another neural circuit model based only on grid cells considers that the recurrent connection weights between grid cells are velocity-dependent [14]. This requires that neuronal connections are instantly tuned by the agent' velocity and is hence unlikely biologically feasible. Our neural circuit model has the similar structure as that in [15, 16], but has different mathematical formulations.

**Model predictions.** This body of work also generates predictions testable in experiments, including: 1) band cells should be organized into clusters based on their orientation and spacing in the brain, analogous to the way grid cells are organized along the dorsal-ventral axis in MEC [33]; 2) band cells in the same module should further be organized into attractor networks, similar to head-direction cells in the thalamic nuclei and postsubiculum [34]; 3) disruption of band cells should lead to deficits in path integration in downstream grid cells, potentially offering insights into the mechanisms underlying navigation impairments observed in the early stages of Alzheimer's disease [35]. Investigating these computationally derived predictions using experiments will not only prompt a re-evaluation of the functional role of grid cells in providing a spatial metric for the environment, but also advance our understanding of the computational contribution of band cells in neural circuits.

**Limitation and future works.** Several important aspects remain to be explored in future studies. For example, it is not fully clear whether undefined cells in the trained RNN hold certain spatial tuning properties, such as those seen in head direction cells [36] or boundary vector cells [37], and whether they play a role in path integration; and if so, at which point along the hierarchical pathway they contribute. Additionally, the nature of interactions between band and grid cells in the trained RNN remains not fully solved, and it needs to be inspected more thoroughly whether these interactions correspond directly to the connectivity profile in our proposed mechanistic neural model. Finally, the current training paradigm is restricted to path integration in physical space. It will be valuable to explore whether this framework is generalizable to path integration in abstract spaces (e.g., social network, value space in decision making) and whether the same hierarchical pathway is preserved [38].

**Cross-talk between AI and neuroscience.** AI approaches are a powerful tool to train network models for executing tasks, but they also face the challenge that the trained networks are often hard to interpret. Without prior knowledge, one has no cue to open up the black box. Here, we demonstrate that by using neuroscience prior knowledge and methods, we can uncover the detailed hierarchical pathway embedded in the trained RNNs for path integration. On the other hand, the RNN-based findings can advance our understanding of the neural circuit mechanism underlying path integration in the brain. This study presents an example of using cross-field knowledge to facilitate AI interpretation and neuroscience research.

## Acknowledgments and Disclosure of Funding

This work was supported by the National Natural Science Foundation of China (no. T2421004 to S.W.), the National Key Research and Development Program of China (2024YFF1206500), the Science and Technology Innovation 2030-Brain Science and Brain-inspired Intelligence Project (no. 2021ZD0200204, S.W.), and the Wellcome Principal Research Fellowship (222457/Z/21/Z, N.B.).

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

# A  Training details of RNN

To train RNNs on path integration tasks, we implemente an established computational framework adapted from the open-source repository developed by Ganguli Lab (`https://github.com/ganguli-lab/grid-pattern-formation`). Code used for this paper is publicly available at `https://github.com/yuling-wu/band_grid_hierarchy`. Vanilla RNN architectures are trained using the hyperparameters specified in Table S1. Our weight matrices are initialized randomly using PyTorch's default scheme and there are no hand-crafted or localized initialization tricks. The trajectory used to train each iteration or test is sampled randomly from a distribution. The training loss is presented in Fig. S1.

Table S1: **Parameters used to train Vanilla RNNs**

| Parameter | Value |
|---|---|
| Arena size | (2.2m x 2.2m) |
| Average agent speed | 0.1 m/sec |
| Place cells ($N_p$) | 512 |
| $\sigma_1$ | 0.12 m |
| $\sigma_2$ | 0.24 m |
| Recurrent units ($N_R$) | 4096 |
| Path length | 20 |
| Epochs | 100 |
| Batch size | 200 |
| Batches per epoch | 1000 |
| Learning rate | $10^{-4}$ |
| l2 regularization | $10^{-4}$ |
| Optimizer | RMSProp |

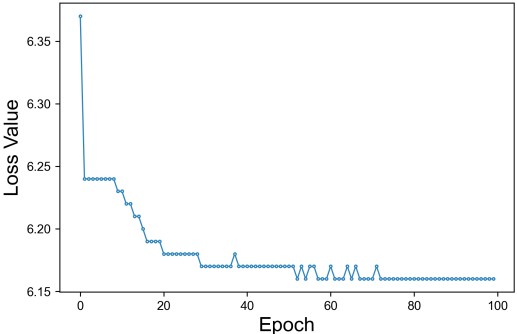

Figure S1:  Training loss for the vanilla RNN.

**Place cell activity:** The receptive field centers of place cells are randomly distributed across the square environment. Each place cell's firing activity is modeled using a difference-of-softmax tuning curve, analogous to the difference-of-Gaussians function:

$$p_i(t) = \text{DoG}[\mathbf{x}(t), \mathbf{c}_i] = \frac{e^{-\|\mathbf{x}-\mathbf{c}_i\|^2/2\sigma_1^2}}{\sum\limits_{j=1}^{N_p} e^{-\|\mathbf{x}-\mathbf{c}_j\|^2/2\sigma_1^2}} - \frac{e^{-\|\mathbf{x}-\mathbf{c}_i\|^2/2\sigma_2^2}}{\sum\limits_{j=1}^{N_p} e^{-\|\mathbf{x}-\mathbf{c}_j\|^2/2\sigma_2^2}} \tag{10}$$

where $\mathbf{x}$ is the current location of the agent, $\mathbf{c}_i$ denotes the center of the $i_{th}$ place cell's receptive field, and $\sigma_1$ and $\sigma_2$ parameterize the width of the center and surround, respectively. The network received velocity inputs derived from the simulated trajectory and was trained to produce the simulated place cell activities as output.

**Path integration error:** Locations of the agent are decoded from the place cell activities by considering three maximally active place cells and averaging the x and y coordinates of their place

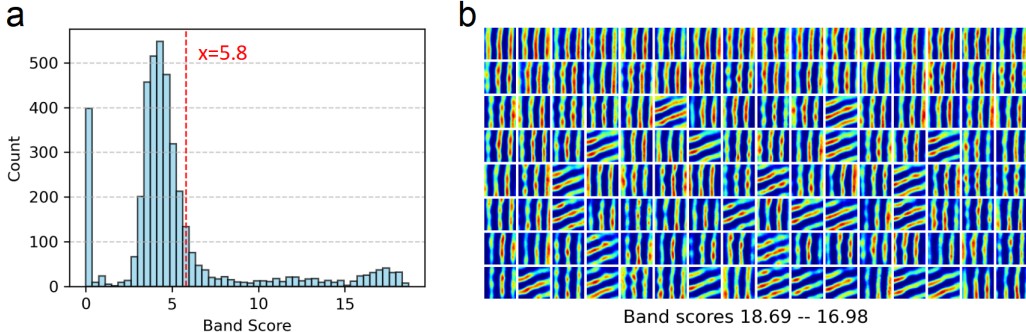

Figure S2: (a) Band score distribution. (b) The top 128 band cells with the highest scores.

field centers. We quantified decoding accuracy using the mean Euclidean distance between the decoded positions and ground truth locations:

$$\mathcal{E} = \frac{1}{N} \sum_{i=1}^{N} \sqrt{(x_i - \hat{x}_i)^2 + (y_i - \hat{y}_i)^2} \tag{11}$$

where $(x_i, y_i)$ denotes the true position and $(\hat{x}_i, \hat{y}_i)$ represents the decoded position for the i-th sample.

## B   Using neuroscience knowledge to analyse RNN

### B.1   Rate map construction

Rate mapping—a fundamental technique in systems neuroscience—quantifies how neuronal firing rates vary with spatial location, providing crucial insights into neural representations of space. When applied to RNNs trained on path integration, this approach has revealed the spontaneous emergence of functionally specialized units analogous to biological grid cells, and band cells observed in biological navigation systems. To do this, we generate a test batch ($n = 200$ trajectories) as inputs to the RNN and record the activation states of all recurrent layer units, the agent's current position and head direction at each step.

To construct spatial rate maps, we partition the environment into an $m \times m$ grid, resulting in $m^2$ spatial bins, and compute the mean activation for each unit within each bin. Here we set $m = 20$ to compute a set of low resolution maps to use for evaluating grid score and set $m = 50$ to compute a set of high resolution maps to plot the resulting tuning curves and visualize the spatial firing patterns. To construct angular activation profiles, we partition the head direction ranging from $-\pi$ to $\pi$ into 100 bins, and compute the mean activation for each unit within each bin.

### B.2   Neuron type classification

**Band cell:** Ideal band cells exhibit a defining electrophysiological signature characterized by spatially periodic firing fields that form bands across the environment [9]. To quantify this, we developed a spectral analysis method. We first compute the 2D Fourier transform of the spatial rate map, suppressing negative frequencies. The resulting power spectrum is fitted with a 2D Gaussian model parameterized by amplitude (A), spatial frequency (k), orientation angle ($\theta$), and bandwidth ($\sigma$) using constrained nonlinear optimization. The band score is calculated as the normalized correlation between the actual spectral power distribution and the Gaussian model, weighted by $\sigma$:

$$\text{Band score} = \frac{\langle \mathbf{F}, \mathbf{G} \rangle}{\|\mathbf{F}\| \cdot \|\mathbf{G}\|} \cdot \frac{1}{\sigma} \tag{12}$$

where:

- **F** is the spectral power matrix obtained from the 2D Fourier transform of the rate map:

$$F(u, v) = |\mathcal{F}\{R(x, y)\}|\big|_{\text{shifted}} \tag{13}$$

with negative frequencies suppressed ($F(u, v) = 0$ for $u < \frac{N}{2}$)

- **G** is the fitted 2D Gaussian model:

$$G(x, y) = A \exp\left(-\frac{(x - k\cos\phi)^2 + (y - k\sin\phi)^2}{2\sigma^2}\right) \quad (14)$$

- Parameters are optimized via:

$$(A^*, k^*, \phi^*, \sigma^*) = \operatorname*{argmin}_{A, k, \phi, \sigma} \|\mathbf{F} - \mathbf{G}\|^2 \quad (15)$$

subject to $k \in [0.2, 1]$, $\phi \in [0, \pi]$, $\sigma \in [0.05, 0.5]$

- The spatial frequency components are derived as:

$$k_x = \frac{k^* \cos\phi^*}{2\Delta x}, \quad k_y = \frac{k^* \sin\phi^*}{2\Delta x} \quad (16)$$

where $\Delta x$ is the spatial bin size.

Band score distribution is show in Fig. S2a and cells with a band score of over 5.8 are classified as band cells. Example band cells are shown in Fig. S2b.

**Grid cell:** Ideal grid cells exhibit a defining electrophysiological signature characterized by spatially periodic firing fields that form a hexagonal lattice tessellation across the environment [4]. Grid score was quantified through a rotational analysis of the autocorrelogram derived from the spatial rate map. Specifically, the autocorrelogram underwent circular rotations in 30° increments, followed by computation of Pearson correlations between each rotated version and the original map. The final grid score metric was calculated as:

$$\text{Grid Score} = \frac{\rho(60°) + \rho(120°)}{2} - \frac{\rho(30°) + \rho(90°) + \rho(150°)}{3} \quad (17)$$

where $\rho(\theta)$ represents the correlation coefficient after $\theta$-degree rotation. This measure captures the hexagonal periodicity signature by comparing correlations at the theoretical peaks (60° and 120°) versus troughs (30°, 90°, and 150°) of an ideal grid pattern. Finally, cells with a grid score of over 0.88 are classified as grid cells.

**Undefined cell:** Any cells that are neither band nor grid cells are classified as undefined cells.

### B.3 Properties of band cells tuning

**Spatial tuning:** To quantitatively characterize band cell activity patterns, we derive three key spatial parameters from the spatial rate maps:

- **Spacing** $\lambda$, the distance between adjacent bands, calculated by $\lambda = 2\pi / \sqrt{k_x^2 + k_y^2}$, where $k_x$ and $k_y$ are the same as in Eq.16;

- **Orientation** $\theta$, the orientation of bands in the 2D space, computed through the following spectral analysis procedure:

$$\theta = \left(\pi - \left[\frac{1}{2}\text{angle}\left(\sum_{\omega} P(\omega)e^{i2\theta(\omega)}\right) + \pi\right] - \frac{\pi}{2}\right) \mod \pi \quad (18)$$

where $\theta(\omega) = \tan^{-1}(f_y/f_x)$ is the orientation of frequency component $\omega$ (constrained to $[0, \pi]$), $P(\omega)$ is the spectral power at frequency $\omega$, the $\frac{1}{2}\text{angle}(\cdot)$ handles the $\pi$-periodicity of band patterns, the $\pi - (\cdot) - \pi/2$ sequence performs the requested axis transformation, and the modulo $\pi$ operation ensures angular continuity;

- **Phase** $\phi$, the positional separation between band cells having the same spacing and orientation. Denote $\Delta \in (-\lambda/2, \lambda/2)$ the offset between the adjacent bands of two band cells, their phase difference is given by $\phi = (\Delta/\lambda) \times \pi \in (-\pi, \pi)$.

As illustrated in Fig. S3a, the connection weight of band cells varies with their phase difference, reminiscent to the center-surround connection pattern.

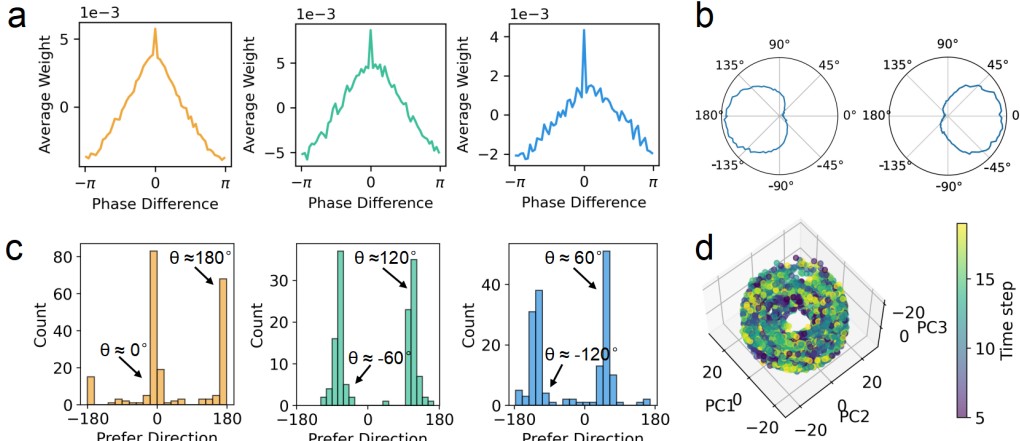

Figure S3: (a) The averaged connection profile of band cells as a function of their phase difference. In figures a and c, colors indicate distinct band cell clusters, and each cluster has the same spacing and orientation. (b) Example band cells with opposite preferred direction. (c) Histograms of preferred directions of band cells in three clusters, which have a bimodal shape peaked at either the cluster orientation $\theta$ or its opposite direction $\theta + \pi$. (d) 3D isomap embedding population activity from a band cell cluster.

**Directional tuning:** To quantitatively characterize band cell direction, we calculate two parameters from the angular activation profiles:

- The **direction score** quantifies directional tuning by fitting angular activation patterns $\mathbf{y}_i(\psi)$ with a circular Gaussian function:

$$\mathbf{y}_{\text{model}}(\psi) = A \exp\left(-\frac{d(\psi, \mu)^2}{2\sigma^2}\right) \qquad (19)$$

where $d(\psi, \mu) = \min(|\psi - \mu|, 2\pi - |\psi - \mu|)$ is the circular distance between angle $\phi$ and preferred direction $\mu$, and $\sigma$ determines tuning width. Parameters $(A, \mu, \sigma)$ are estimated via nonlinear least-squares fitting initialized at $[1, 0, 1]$. The direction score is computed as the normalized projection between observed ($\mathbf{y}_{\text{data}}$) and fitted ($\mathbf{y}_{\text{model}}$) activation patterns:

$$\text{Direction Score} = \frac{\mathbf{y}_{\text{data}} \cdot \mathbf{y}_{\text{model}}}{\|\mathbf{y}_{\text{data}}\| \|\mathbf{y}_{\text{model}}\|} \qquad (20)$$

- The **preferred direction** $\xi_i \in [-\pi, \pi)$ for each band cell $i$ is derived from its angular activation profile $y_i(\psi_k)$ through circular moment analysis, computed as:

$$\xi_i = \arg\left(\sum_{k=1}^{N_\psi} y_i(\psi_k) e^{i\psi_k}\right) \qquad (21)$$

where $\psi_k = -\pi + 2\pi k/N_\psi$ $(k = 0, ..., N_\psi - 1)$ defines $N_\psi$ equally spaced angular bins, $e^{i\psi_k}$ represents the unit vector in complex space, and $\arg(\cdot)$ extracts the phase of the resultant vector.

We find that, for each cluster with the same spacing and direction tuning, the majority of band cells form two subgroups with opposite preferred directions (Fig. S3b-c).

## B.4 Population activity analysis

Neural population dynamics were analyzed using the following pipeline:

- **Data extraction**: A test batch ($n = 200$ trajectories) was generated as inputs to the RNN to generate the network's corresponding latent representations. Then neuron subset $\mathbf{X}_{\text{raw}}$ was extracted (shape: $T \times N$, where $T$ is timepoints and $N$ is neurons to be analyzed).

- **Preprocessing**: Neural activity was standardized using z-score normalization:

$$\mathbf{X}_{\text{scaled}} = \text{scaler}(\mathbf{X}_{\text{raw}}) \tag{22}$$

- **Dimensionality reduction**:
  - PCA projected data to 15 principal components
  - Isomap further reduced to 3D ($k = 8$ neighbors, geodesic distance metric)

$$\mathbf{X}_{\text{lowdim}} = \text{Isomap}(\text{PCA}(\mathbf{X}_{\text{scaled}})) \tag{23}$$

- **Visualization**: The 3D neural manifold was plotted against animal trajectory (5-frame offset corrected) using time-indexed coloring (viridis colormap).

  Fig. S3d displays an alternative Isomap embedding for a distinct band cell group not analyzed in the main text.

## C   Dissecting information flows in RNN

### C.1   Pruning experiments

To systematically investigate the computational roles of recurrent layer units in path integration, we performed four distinct pruning experiments with rigorous controls:

- **Pruning velocity inputs**: For each pruning level $n \in \{1, \ldots, 700\}$, we randomly selecte $n$ units across cell types and nullify their velocity input weights:

$$\mathbf{M}_{i,\cdot}^{(n)} \leftarrow \mathbf{0}, \quad i \in \mathcal{S}_n$$

  where $\mathcal{S}_n$ denotes a size-$n$ random subset of units, and $\mathbf{M} \in \mathbb{R}^{N_R \times 2}$ encodes velocity-to-recurrent layer weights.

- **Pruning read-out connections**: We prune output projections from $n$ randomly selected units by zeroing their readout weights:

$$\mathbf{W}_{\cdot,j}^{(n)} \leftarrow \mathbf{0}, \quad j \in \mathcal{S}_n$$

  with $\mathbf{W} \in \mathbb{R}^{N_P \times N_R}$ representing the recurrent-to-output weight matrix.

- **Pruning cell types**: For comprehensive cell inactivation, we simultaneously nullified all incoming and outgoing connections of selected units:

$$\mathbf{J}_{i,\cdot} \leftarrow \mathbf{0}, \quad \mathbf{J}_{\cdot,i} \leftarrow \mathbf{0}, \quad \forall i \in \mathcal{S}_n$$

  where $\mathbf{J} \in \mathbb{R}^{N_R \times N_R}$ is the recurrent weight matrix.

- **Pruning connections between band and grid cells**:

  For the cell-type specific pruning analysis, we conducted bidirectional connection pruning between band (A) and grid (B) cell populations:

  - **A→B Pruning**: Nullified all forward connections from a random subset of $n$ band cells to *every* grid cell:

$$\mathbf{J}_{A_k, B_j}^{(n)} \leftarrow 0, \quad \forall k \in \mathcal{S}_n, \forall j \in \{1, ..., N_B\}$$

  - **B→A Pruning**: Symmetrically ablated all feedback connections from a random subset of $n$ grid cells to *every* band cell:

$$\mathbf{J}_{B_k, A_j}^{(n)} \leftarrow 0, \quad \forall k \in \mathcal{S}_n, \forall j \in \{1, ..., N_A\}$$

**Error bars** represent $\pm$ standard deviation across $n = 30$ independent repetitions, reflecting the variability in individual measurements.

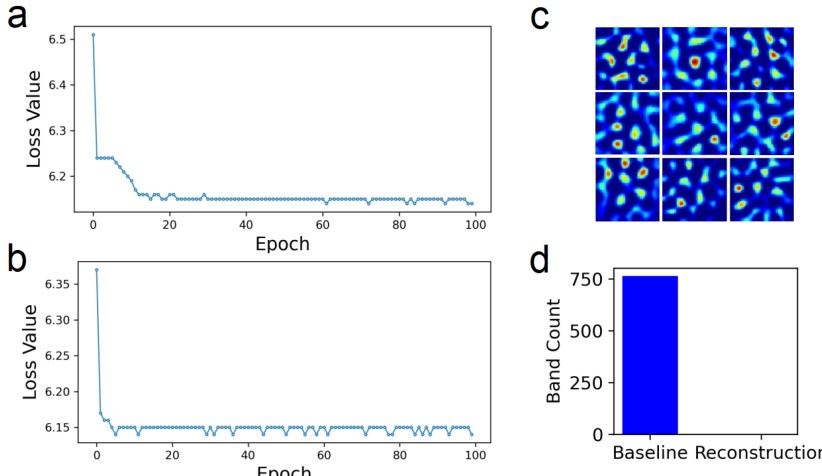

Figure S4: (a) Training loss for the RNN with predefined hierarchical structure. (b) Training loss for RNN for location reconstruction. (c) Example grid cells found in the newly trained RNN for location reconstruction. (d) The number of band cells found in the Vanilla RNN (Baseline) and newly trained RNN (Reconstruction).

## C.2 RNN with predefined hierarchical structure

We implement a hierarchical RNN architecture that takes the hierarchical pathway as a prior (Fig. 4b). The model consists of two reciprocally connected modules:

- **First module (RNN1)**: Processes velocity inputs ($\mathbf{v}_t$) combined with feedback from the second module, designed to emulate the properties of band cells:

$$\mathbf{r}_{t+1}^{(1)} = \sigma\left(\mathbf{J}_1\mathbf{r}_t^{(1)} + \mathbf{J}_2\mathbf{r}_t^{(2)} + \mathbf{M}\mathbf{v}_t\right) \tag{24}$$

  where $\mathbf{r}_t^{(2)}$ represents the feedback from RNN2, $\mathbf{J}_1$ and $\mathbf{J}_2$ represent the recurrent weights of RNN1 and RNN2 respectively.

- **Second module (RNN2)**: Receives inputs from RNN1 and generates spatial representations, mimicking grid cell functionality:

$$\mathbf{r}_{t+1}^{(2)} = \sigma(\mathbf{J}_1\mathbf{r}_{t+1}^{(1)} + \mathbf{J}_2\mathbf{r}_t^{(2)}) \tag{25}$$

We train this hierarchical RNN model on the standard path integration task using the same hyperparameters specified in Table S1. The training loss is presented in Fig. S4 a.

## C.3 RNN trained for location reconstruction

We modify the task from path integration to pure location reconstruction by replacing velocity inputs of the vanilla RNN with place cell activations at each timestep, expressed as:

$$\mathbf{r}_{t+1} = \sigma\left(\mathbf{J}\mathbf{r}_t + \mathbf{M}\mathbf{p}_t\right) \tag{26}$$

where $\mathbf{p}_t \in \mathbb{R}^{N_g}$ denotes the ground truth place cell activation pattern corresponding to the current position. All hyperparameters are the same as in Table S1. The training loss is presented in Fig. S4 b. After training the RNN on this modified task, we observe the emergence of grid-like activity patterns while notably lacking band cell representations (Fig. S4 c-d).

# D   Quantitative Performance Analysis

## D.1   Error Distribution Analysis

We quantified path integration performance across 30 independent trials for each of 10 different RNN implementations (Seeds 0–9). The results demonstrate consistent performance, with errors (normalized by arena size) reported as mean $\pm$ standard deviation (units: %):

$$\text{Seed } 0 : 2.09 \pm 0.06 \quad \text{Seed } 5 : 2.07 \pm 0.04$$
$$\text{Seed } 1 : 2.09 \pm 0.05 \quad \text{Seed } 6 : 2.08 \pm 0.05$$
$$\text{Seed } 2 : 2.08 \pm 0.05 \quad \text{Seed } 7 : 2.09 \pm 0.06$$
$$\text{Seed } 3 : 2.07 \pm 0.05 \quad \text{Seed } 8 : 2.13 \pm 0.06$$
$$\text{Seed } 4 : 2.12 \pm 0.07 \quad \text{Seed } 9 : 2.11 \pm 0.07$$

## D.2   Computational Complexity Scaling

We analyzed the relationship between hidden layer size ($N_R$) and path integration error, with errors (normalized by arena size) reported as mean $\pm$ standard deviation (units: %):

$$N_R = 1024 : 2.57 \pm 0.14$$
$$N_R = 2048 : 2.30 \pm 0.08$$
$$N_R = 4096 : 2.09 \pm 0.06$$
$$N_R = 8192 : 2.06 \pm 0.05$$

This analysis demonstrates that our model achieves stable performance ($< 2.19\%$ error) with $N \geq 4096$ neurons, suggesting our approach provides computationally efficient spatial representation.

## D.3   Pruning Baseline

For each pruning condition, we prune connections selected from random groups as baseline to assess the relative importance of our specific pruning choices. The key results are presented in Table S2, which demonstrates that pruning specific functional connections (input-to-band or grid-read-out) produces significantly larger errors compared to random pruning, suggesting specialized functional roles for these specific cell groups.

Table S2: Path integration error increase (mean $\pm$ SD across 30 trials when 750 connections are pruned). Error is measured by the standard deviation of location prediction errors divided by the arena size (unit: %).

| Pruning Type | Band cell | Grid cell | Undefined cell | Random cell |
|---|---|---|---|---|
| Pruning velocity inputs | $2.32 \pm 0.43$ | $0.11 \pm 0.07$ | $0.14 \pm 0.09$ | $0.46 \pm 0.07$ |
| Pruning read-out connections | $2.67 \pm 0.50$ | $9.15 \pm 0.71$ | $0.30 \pm 0.11$ | $0.81 \pm 0.22$ |

# E   Reproducibility

## E.1   Experiments Across Different Seeds

we repeat the experiments across 10 random seeds for Vanilla RNN. Our results robustly demonstrate that band cells consistently serve as the primary recipients of velocity inputs and grid cells reliably encode spatial location information. The additional results are:

Neuron Counts and Pruned Connections:

| Seed | Band Cells | Grid Cells | Undefined Cells | Pruned Connections |
|------|-----------|-----------|-----------------|--------------------|
| 0 | 868 | 443 | 2785 | 443 |
| 1 | 818 | 526 | 2752 | 526 |
| 2 | 914 | 267 | 2915 | 267 |
| 3 | 823 | 146 | 3127 | 146 |
| 4 | 782 | 457 | 2857 | 457 |
| 5 | 854 | 442 | 2800 | 442 |
| 6 | 972 | 228 | 2896 | 228 |
| 7 | 535 | 324 | 3237 | 324 |
| 8 | 553 | 597 | 2946 | 553 |
| 9 | 810 | 394 | 2892 | 394 |

Pruning Performance (mean $\pm$ SD):

Velocity Input Pruning:

| Seed | Band | Grid | Undefined |
|------|------|------|-----------|
| 0 | $1.8282 \pm 0.1026$ | $0.0003 \pm 0.1069$ | $0.0106 \pm 0.0990$ |
| 1 | $2.5350 \pm 0.1058$ | $0.0401 \pm 0.0616$ | $0.0475 \pm 0.0720$ |
| 2 | $0.7015 \pm 0.0786$ | $0.0247 \pm 0.0687$ | $0.0248 \pm 0.0736$ |
| 3 | $0.3086 \pm 0.0679$ | $0.0022 \pm 0.0744$ | $0.0254 \pm 0.0701$ |
| 4 | $2.2449 \pm 0.1073$ | $0.0430 \pm 0.0955$ | $0.0173 \pm 0.0710$ |
| 5 | $2.0076 \pm 0.0884$ | $0.0217 \pm 0.0480$ | $0.0172 \pm 0.0733$ |
| 6 | $0.6119 \pm 0.0963$ | $0.0123 \pm 0.0683$ | $0.0003 \pm 0.0732$ |
| 7 | $1.2673 \pm 0.0806$ | $0.0242 \pm 0.0697$ | $0.0603 \pm 0.0803$ |
| 8 | $2.7330 \pm 0.1191$ | $0.0411 \pm 0.0625$ | $0.2117 \pm 0.0608$ |
| 9 | $1.3750 \pm 0.1083$ | $0.0124 \pm 0.0815$ | $0.0225 \pm 0.0795$ |

Read-out Connection Pruning:

| Seed | Band | Grid | Undefined |
|------|------|------|-----------|
| 0 | $0.9446 \pm 0.2144$ | $4.2009 \pm 0.6104$ | $0.1558 \pm 0.1297$ |
| 1 | $1.7184 \pm 0.2905$ | $7.4244 \pm 0.7040$ | $0.4433 \pm 0.1778$ |
| 2 | $0.4024 \pm 0.1543$ | $0.7765 \pm 0.1850$ | $0.1569 \pm 0.1119$ |
| 3 | $0.1272 \pm 0.0737$ | $0.2362 \pm 0.1024$ | $0.1058 \pm 0.0894$ |
| 4 | $0.5466 \pm 0.1479$ | $1.6829 \pm 0.2364$ | $0.2440 \pm 0.1109$ |
| 5 | $0.7177 \pm 0.2362$ | $2.3296 \pm 0.3476$ | $0.1877 \pm 0.1093$ |
| 6 | $0.2365 \pm 0.1038$ | $0.7850 \pm 0.2799$ | $0.0657 \pm 0.0990$ |
| 7 | $0.3730 \pm 0.1556$ | $1.0181 \pm 0.2656$ | $0.0985 \pm 0.0967$ |
| 8 | $2.6436 \pm 0.3403$ | $8.3318 \pm 0.6056$ | $0.2729 \pm 0.1016$ |
| 9 | $0.6141 \pm 0.1355$ | $2.8032 \pm 0.5064$ | $0.1566 \pm 0.1104$ |

### E.2 Different Training Paragram

To check the flexibility of our results, we trained three additional path-integration RNN variants:

- **Vanilla RNN $\rightarrow$ LSTM transition** (different in net structure).
- **Xu et al. (2022) [19] training paradigm** (different in net structure (RNN and LSTM), place field (Gaussian), and loss function (conformal isometry and path integration loss)).
- **Petterson et al. (2024) [24] training paradigm** (different in initial state (MLP), loss function (distance preservation and capacity loss), and decoder (none)).

The quantitative analysis of path integration performance across the three RNN variants are presented below, which demonstrates that velocity inputs are preferentially processed by band cells across all three RNN variants, as evidenced by their greater sensitivity to velocity pruning. However, the absence of decoder modules in some variants prevented systematic pruning read-out experiments, limiting direct comparisons of read-out performance.

### E.2.1 Vanilla RNN→LSTM transition

The pruning experiments show that models degrade severely when velocity inputs to band cells are pruned, but remain relatively stable when inputs to grid cells or undefined cells are pruned under the same conditions:

Neuron Counts and Pruned Connections:

| Seed | Band Cells | Grid Cells | Undefined Cells | Pruned Connections |
|---|---|---|---|---|
| 0 | 346 | 534 | 3216 | 346 |
| 1 | 258 | 276 | 3562 | 258 |
| 2 | 524 | 554 | 3018 | 524 |
| 3 | 394 | 305 | 3397 | 305 |
| 4 | 500 | 186 | 3410 | 186 |
| 5 | 313 | 136 | 3647 | 136 |
| 6 | 294 | 57 | 3745 | 57 |
| 7 | 324 | 225 | 3547 | 225 |
| 8 | 258 | 298 | 3540 | 258 |
| 9 | 363 | 124 | 3609 | 124 |

Velocity Input Pruning Performance (mean $\pm$ SD):

| Seed | Band Cells | Grid Cells | Undefined Cells |
|---|---|---|---|
| 0 | $1.3948 \pm 0.1030$ | $0.0070 \pm 0.0667$ | $0.0809 \pm 0.0785$ |
| 1 | $1.3914 \pm 0.1392$ | $0.0261 \pm 0.0729$ | $0.0708 \pm 0.0718$ |
| 2 | $1.8552 \pm 0.1261$ | $0.0638 \pm 0.0974$ | $0.1554 \pm 0.1085$ |
| 3 | $1.1153 \pm 0.1696$ | $0.0715 \pm 0.1594$ | $0.0511 \pm 0.1202$ |
| 4 | $0.2639 \pm 0.0781$ | $0.0128 \pm 0.0605$ | $0.0268 \pm 0.0613$ |
| 5 | $0.3274 \pm 0.0774$ | $0.0150 \pm 0.0647$ | $0.0046 \pm 0.0776$ |
| 6 | $0.0811 \pm 0.0649$ | $0.0067 \pm 0.0548$ | $0.0036 \pm 0.0597$ |
| 7 | $0.8140 \pm 0.1018$ | $0.0022 \pm 0.0689$ | $0.0610 \pm 0.0805$ |
| 8 | $0.7389 \pm 0.0691$ | $0.0303 \pm 0.0568$ | $0.0905 \pm 0.0544$ |
| 9 | $0.2011 \pm 0.1214$ | $0.0223 \pm 0.1245$ | $0.0191 \pm 0.1048$ |

### E.2.2 Xu et al. (2022)

The pruning experiments reveal that band cells exhibit significantly greater sensitivity to input manipulation compared to grid or undefined cells, which is consistently observed across all experimental seeds except seed 1:

Neuron Counts and Pruned Connections:

| Seed | Band Cells | Grid Cells | Undefined Cells | Pruned Connections |
|---|---|---|---|---|
| 0 | 347 | 403 | 1050 | 347 |
| 1 | 313 | 424 | 1063 | 313 |
| 2 | 303 | 363 | 1134 | 303 |
| 3 | 316 | 369 | 1115 | 316 |
| 4 | 312 | 384 | 1104 | 312 |
| 5 | 311 | 382 | 1107 | 311 |
| 6 | 318 | 427 | 1055 | 318 |
| 7 | 302 | 452 | 1046 | 302 |
| 8 | 318 | 418 | 1064 | 318 |
| 9 | 312 | 401 | 1087 | 312 |

Velocity Input Pruning Performance (mean $\pm$ SD):

| Seed | Band Cells | Grid Cells | Undefined Cells |
|---|---|---|---|
| 0 | $8.3573 \pm 1.3764$ | $0.3688 \pm 0.2071$ | $2.9723 \pm 1.1574$ |
| 1 | $9.6032 \pm 1.7588$ | $0.0395 \pm 0.2031$ | $10.0732 \pm 1.8683$ |
| 2 | $13.5328 \pm 2.1336$ | $0.0767 \pm 0.1197$ | $4.0376 \pm 0.6832$ |
| 3 | $14.8651 \pm 1.8007$ | $0.8835 \pm 0.2180$ | $4.9369 \pm 0.8729$ |
| 4 | $15.2718 \pm 2.3267$ | $-0.0064 \pm 0.0720$ | $2.6171 \pm 0.4573$ |
| 5 | $10.3343 \pm 2.0101$ | $0.3419 \pm 0.3565$ | $4.1531 \pm 0.7071$ |
| 6 | $9.5627 \pm 1.3337$ | $-0.0611 \pm 0.1900$ | $5.5746 \pm 0.9934$ |
| 7 | $9.3729 \pm 1.6383$ | $0.2143 \pm 0.3917$ | $5.8331 \pm 1.1967$ |
| 8 | $11.2179 \pm 1.9553$ | $3.3365 \pm 1.3821$ | $4.1106 \pm 0.7282$ |
| 9 | $5.1514 \pm 1.3389$ | $-0.6553 \pm 0.8521$ | $3.9936 \pm 1.6025$ |

### E.2.3 Petterson et al. (2024)

Our 10-seed replication of Petterson et al. (2024) demonstrates robust preferential processing of velocity inputs by band cells. The additional results are:

Neuron Counts and Pruned Connections:

| Seed | Band | Grid | Undefined | Pruned Connections |
|---|---|---|---|---|
| 0 | 37 | 196 | 23 | 23 |
| 1 | 26 | 200 | 30 | 26 |
| 2 | 26 | 209 | 21 | 21 |
| 3 | 38 | 194 | 24 | 24 |
| 4 | 26 | 194 | 36 | 26 |
| 5 | 38 | 203 | 15 | 15 |
| 6 | 24 | 202 | 30 | 24 |
| 7 | 35 | 201 | 20 | 20 |
| 8 | 34 | 205 | 17 | 17 |
| 9 | 36 | 202 | 18 | 18 |

Pruning Performance (mean $\pm$ SD):

| Seed | Band | Grid | Undefined |
|---|---|---|---|
| 0 | $2.5472 \pm 0.3644$ | $0.0368 \pm 0.0278$ | $0.6565 \pm 0.0061$ |
| 1 | $5.2265 \pm 0.0205$ | $0.0198 \pm 0.0121$ | $0.7424 \pm 0.1176$ |
| 2 | $3.4828 \pm 0.2735$ | $0.0297 \pm 0.0187$ | $1.2256 \pm 0.0086$ |
| 3 | $3.1603 \pm 0.3524$ | $0.0147 \pm 0.0075$ | $0.6654 \pm 0.0072$ |
| 4 | $3.3173 \pm 0.0183$ | $0.0315 \pm 0.0250$ | $1.5762 \pm 0.3514$ |
| 5 | $1.8896 \pm 0.2988$ | $0.0122 \pm 0.0071$ | $0.1567 \pm 0.0012$ |
| 6 | $3.6600 \pm 0.0223$ | $0.0223 \pm 0.0114$ | $1.5868 \pm 0.2246$ |
| 7 | $2.8943 \pm 0.3955$ | $0.0117 \pm 0.0074$ | $0.4566 \pm 0.0030$ |
| 8 | $2.5261 \pm 0.3254$ | $0.0147 \pm 0.0099$ | $0.3661 \pm 0.0035$ |
| 9 | $2.3297 \pm 0.3849$ | $0.0205 \pm 0.0176$ | $0.4389 \pm 0.0034$ |

## F   Neural circuit model details

### F.1   Overview of the model architecture

We provide a mathematical overview of the neural circuit model for 2D path integration, consisting of band cell modules and a grid cell module. Each component performs specific computations described below.

**1. Pure band cell dynamics (1D CANN)**

Each band cell module maintains a bump of activity along a 1D phase space $\phi \in (-\pi, \pi]$, via recurrent dynamics:

$$\tau \frac{\partial u_b(\phi, t)}{\partial t} = -u_b(\phi, t) + \rho \int J_b(\phi, \phi') \, r_b(\phi', t) \, d\phi' + \rho \sum_{m=\pm} \int W_m(\phi, \phi') \, r_m(\phi', t) \, d\phi' + I_g(\phi, t),$$

$$(27)$$

where $u_b$ and $r_b$ are synaptic input and firing rate of pure band cells. The recurrent connectivity is symmetric and translation-invariant:

$$J_b(\phi, \phi') = \frac{J_b^0}{\sqrt{2\pi}\sigma_b} \exp\left[-\frac{|\phi - \phi'|_\pi^2}{2\sigma_b^2}\right]. \tag{28}$$

Their firing rates are normalized to stabilize the bump:

$$r_b(\phi, t) = \frac{u_b^2(\phi, t)}{1 + k_b\rho \int u_b^2(\phi', t)\, d\phi'}. \tag{29}$$

## 2. Direction-conjunctive band cells

These cells receive excitatory input from pure band cells and encode the agent's projected velocity along the module's preferred orientation $\theta$ and its opposite direction $\theta + \pi$. Their dynamics are defined as:

$$\tau\frac{\partial u_\pm(\phi, t)}{\partial t} = -u_\pm(\phi, t) + w_b \cdot r_b(\phi, t), \tag{30}$$

$$r_\pm(\phi, t) = [g_0 + (\mathbf{v}(t) \cdot \mathbf{k})u_\pm(\phi, t)]_+ , \tag{31}$$

where $u_\pm(\phi, t)$ and $r_\pm(\phi, t)$ are the synaptic input and firing rate of the conjunctive cells. The velocity projection vector is $\mathbf{k} = (\sin\theta/\lambda, \cos\theta/\lambda)$, determining the speed component along the module's axis. The firing rates reflect the movement direction and strength of the agent.

*Feedback to pure band cells.* Direction-conjunctive cells modulate the activity bump in pure band cells via asymmetric projections. The connection weights from conjunctive to pure band cells are:

$$W_\pm(\phi, \phi') = \frac{W_b^0}{\sqrt{2\pi}\sigma_b} \exp\left[-\frac{|\phi - \phi' \mp \delta|_\pi^2}{2\sigma_b^2}\right]. \tag{32}$$

These connections are offset by $\pm\delta$ along the phase axis and thus induce a translation of the activity bump, driving 1D path integration along the module orientation. Specifically, $W_+$ promotes bump motion in the forward direction ($\theta$), and $W_-$ in the opposite direction ($\theta + \pi$).

## 3. Grid cells

Grid cells are organized on a 2D toroidal manifold and integrate inputs from two band cell modules with orientations $\theta_1$ and $\theta_2$, which are $60°$ apart. They form a 2D continuous attractor neural network (CANN) with the following dynamics:

$$\tau_g\frac{\partial u_g(\boldsymbol{\phi}, t)}{\partial t} = -u_g(\boldsymbol{\phi}, t) + \iint_{-\pi}^\pi J_g(\boldsymbol{\phi}, \boldsymbol{\phi}')\, r_g(\boldsymbol{\phi}', t)\, d\boldsymbol{\phi}' + I_1(\phi_1, t) + I_2(\phi_2, t), \tag{33}$$

where $u_g(\boldsymbol{\phi}, t)$ and $r_g(\boldsymbol{\phi}, t)$ are the synaptic input and firing rate of a grid cell at phase $\boldsymbol{\phi} = (\phi_1, \phi_2)$. The recurrent connectivity $J_g$ is translation-invariant and shaped to support hexagonal grid patterns:

$$J_g(\boldsymbol{\phi}, \boldsymbol{\phi}') = \frac{J_0^g}{2\pi\sigma_g^2} \exp\left[-\frac{||\boldsymbol{\phi} - \boldsymbol{\phi}'||_g^2}{2\sigma_g^2}\right], \tag{34}$$

with firing rates given by a divisive normalization:

$$r_g(\boldsymbol{\phi}, t) = \frac{u_g^2(\boldsymbol{\phi}, t)}{1 + k_g \iint u_g^2(\boldsymbol{\phi}', t)\, d\boldsymbol{\phi}'}. \tag{35}$$

The distance metric $|| \cdot ||_g$ between grid cell phases is defined to produce a hexagonal pattern. Specifically, letting $\Delta\boldsymbol{\phi} = \boldsymbol{\phi} - \boldsymbol{\phi}'$, we apply periodic wrapping and a linear transformation:

$$\Delta\phi_1 = \mathrm{wrap}(\phi_1 - \phi_1'), \tag{36}$$

$$\Delta\phi_2 = \mathrm{wrap}(\phi_2 - \phi_2'), \tag{37}$$

$$\delta_x = \Delta\phi_1, \tag{38}$$

$$\delta_y = \frac{2}{\sqrt{3}}\left(\Delta\phi_2 - \frac{1}{2}\Delta\phi_1\right), \tag{39}$$

$$||\boldsymbol{\phi} - \boldsymbol{\phi}'||_g = \sqrt{\delta_x^2 + \delta_y^2}, \tag{40}$$

where $\mathrm{wrap}(\cdot)$ maps values to $(-\pi, \pi]$ to account for periodic boundary conditions. The linear transformation aligns the phase coordinates with the axes of a hexagonal lattice.

This design ensures that the bump activity on the grid cell manifold has a hexagonal structure rather than a square one, consistent with biological grid cell tuning.

**4. Reciprocal connections between grid and band modules**

Bidirectional connections allow information exchange between grid and band cells.

*From grid cells to band cells:*

$$I_g(\phi_k^b, t) = \iint W_{gb}(\phi_k^b, \boldsymbol{\phi}') \, r_g(\boldsymbol{\phi}', t) \, d\boldsymbol{\phi}', \quad k = 1, 2 \tag{41}$$

*From band cells to grid cells:*

$$I_k(\phi_k, t) = \int W_{gb}(\boldsymbol{\phi}, \phi') \, r_b(\phi', t) \, d\phi', \quad k = 1, 2 \tag{42}$$

The weights are symmetric and phase-aligned:

$$W_{gb}(\phi_k^b, \boldsymbol{\phi}) = \frac{W_{gb}^0}{\sqrt{2\pi}\sigma_{gb}} \exp\left[ -\frac{|\phi_k^b - \phi_k^g|_\pi^2}{2\sigma_{gb}^2} \right]. \tag{43}$$

## F.2 Parameters

The model is composed of multiple *path integration modules*, each responsible for encoding spatial location at a specific scale. Each module consists of two **band cell modules** and one **grid cell module**. The two band modules within a path integration module differ only in their preferred orientation (offset by $60°$), but share all other dynamic and connectivity parameters. The five modules operate at distinct spatial scales $\lambda^{(i)}$, which influence their velocity projection vectors $\mathbf{k}^{(i)}$ and resulting grid periodicity.

Within each module, band cells perform velocity integration using directionally tuned input, coupled through asymmetric recurrent and cross-module connections. Their outputs drive the activity of grid cells, which integrate spatial phase across the two orientations to generate hexagonal grid patterns. The grid cell dynamics are shaped by their own recurrent connectivity and normalization mechanisms.

All modules share a common set of parameters for band and grid dynamics, except for the spatial scale $\lambda$ and the orientation $\theta$, which define the velocity-to-phase projection. These design choices allow for a consistent multi-scale encoding of space, which is later used for spatial decoding via a grid-to-place projection.

The table below summarizes all parameters used in the model and their hierarchical roles.

## F.3 Simulation details

We performed two simulations to evaluate the functionality of the proposed neural circuit model for path integration.

**Simulation 1: Activity mapping in full environment.** We simulated a freely foraging rat in a square box environment, generating a trajectory of 100,000 time steps with a fixed time step size $\Delta t = 0.05$. The simulated trajectory densely covers the entire spatial environment. At each time step, we recorded the activity of different neural populations in the model, including band cells (from each module) and grid cells. We then computed the average firing rate at each spatial location to obtain the spatial tuning maps (i.e., heatmaps) for different cells.

**Simulation 2: Decoding trajectory from neural activity.** In this simulation, we generated a continuous trajectory of 1,000 time steps, again with $\Delta t = 0.05$. Using the spatial tuning maps (heatmaps) of grid cells obtained from Simulation 1, we decoded the animal's position at each time step from the instantaneous population activity of grid cells. The decoding procedure was carried out in two stages:

Table S3: Parameters of the path integration model. Each path integration module consists of two band cell modules (offset by $60°$ in orientation) and one grid cell module, with shared dynamics and connectivity parameters.

| Symbol | Description | Value |
|---|---|---|
| **Global Parameters** | | |
| $\Delta t$ | Simulation time step | 5 ms |
| $n_\lambda$ | Number of path integration modules (scales) | 5 |
| $\lambda^{(i)}$ | Spatial scale of module $i$ | $2.5 + 0.3i$ |
| $\theta^{(1)}, \theta^{(2)}$ | Orientation of the two band cell modules | $0°, 60°$ |
| **Band Cell Modules (per module, shared across scales)** | | |
| $\tau$ | Time constant of pure band cell dynamics | 100 ms |
| $\tau$ | Time constant of conjunctive band cell dynamics | 10 ms |
| $N_b$ | Neuron number of pure band cells | 180 |
| $N_+$ | Neuron number of $v^+$ band cells | 180 |
| $N_-$ | Neuron number of $v^-$ band cells | 180 |
| $J_b^0$ | Strength of recurrent excitation in pure band cells | 1.1 |
| $\sigma_b$ | Width of Gaussian kernel for recurrent/feedforward connections | $\frac{2}{9}\pi$ |
| $k_b$ | Inhibition strength in divisive normalization | $5 * 10^{-4}$ |
| $w_b$ | Strength of pure-to-conjunctive band cell connections | 1 |
| $W_b^0$ | Strength of conjunctive-to-pure feedback connections | 0.2 |
| $\delta$ | Offset in phase space from direction-specific input $v^\pm$ | 0.265 |
| $g_0$ | Baseline activity level of conjunctive band cells | 0.2 |
| **Grid Cell Module (per module, shared across scales)** | | |
| $\tau_g$ | Time constant of grid cell dynamics | 10ms |
| $J_0^g$ | Strength of recurrent connectivity among grid cells | 1 |
| $\sigma_g$ | Width of grid cell recurrent kernel | $\frac{1}{9}\pi$ |
| $k_g$ | Inhibition strength in grid cell divisive normalization | $5 * 10^{-3}$ |
| $W_{gb}^0$ | Strength of connection between band and grid cells | 0.1 |
| $\sigma_{gb}$ | Width of Gaussian kernel for band-to-grid connections | $\frac{2}{9}\pi$ |

1. **Grid-to-place cell transformation:** The grid cell activity vector at each time step was linearly projected onto a population of place cells. Each place cell was assigned a spatial location $(x, y)$ on a uniform 2D grid. To define the synaptic connection between a grid cell and a place cell, we first projected the place cell's spatial location into the phase space $(\phi_1, \phi_2)$ of the two band cell modules used to construct grid cells. This projection was done by computing the inner product of the place field center $(x, y)$ with the two module wave vectors $\mathbf{k}_1$ and $\mathbf{k}_2$, i.e.,

$$\phi_k = 2\pi \cdot \text{mod}\left(\mathbf{k}_k \cdot (x, y), 1\right) - \pi, \qquad k = 1, 2.$$

This yields the wrapped phases $(\phi_1, \phi_2)$ corresponding to the grid cell's toroidal phase space. The connection strength between each grid cell (with preferred phase $\phi^g$) and a place cell (located at $(x, y)$ with corresponding $\phi^{\text{pc}} = (\phi_1, \phi_2)$) is then defined by a Gaussian kernel on the torus:

$$W_{\text{pc},g}(\phi^{\text{pc}}, \phi^g) = \frac{W_0^{\text{pc}}}{2\pi\sigma_{\text{pc}}^2} \exp\left(-\frac{\|\phi^{\text{pc}} - \phi^g\|_g^2}{2\sigma_{\text{pc}}^2}\right),$$

where the toroidal phase distance $\|\cdot\|_g$ is computed using the same hexagonal metric as in the grid cell CANN. Specifically, for $\boldsymbol{d} = \phi^{\text{pc}} - \phi^g$ (wrapped to $[-\pi, \pi]$),

$$\|\boldsymbol{d}\|_g = \sqrt{\delta_x^2 + \left(\frac{\delta_y - \frac{1}{2}\delta_x}{\sqrt{3}/2}\right)^2},$$

with $\delta_x = d_1$ and $\delta_y = d_2$.

This construction ensures that the grid-to-place connection matrix respects the periodic and hexagonal structure of the underlying grid cell code.

2. **Population vector decoding:** After obtaining the activity of place cells from the grid cell input, we decoded the position by computing a population vector (center-of-mass) over place field centers, weighted by the activity of each place cell at each time step.

We then evaluated the model's path integration accuracy by comparing the decoded positions with the true trajectory. Results obtained under idealized conditions—where both the input signals and the model were free of random noise—are presented in Fig. 5. In more realistic noisy scenarios, small errors accumulate over time, resulting in a gradual drift in the estimated position, as illustrated in Fig. S5.

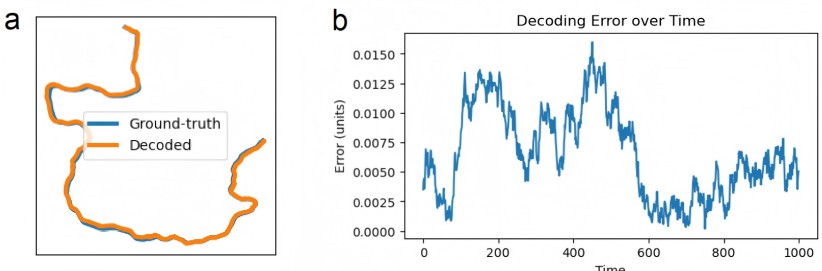

Figure S5: (a) Inferred trajectory of the noisy neural circuit model compared with the ground-truth trajectory. (b) Decoding error of the neural circuit under noisy conditions.

