# OpenReview forum: "Unfolding the Black Box of Recurrent Neural Networks for Path Integration"
_NeurIPS.cc/2025/Conference — NeurIPS 2025 poster_

### Official Review · Reviewer_pJcG · 2025-06-29

**Clarity:** 3
**Significance:** 3
**Originality:** 3
**Rating:** 4
**Confidence:** 4

**Summary:**

Previous studies have implicated various neural substrates, especially the medial entorhinal cortex (MEC), in path integration. While previous RNN-based models showed that networks trained for path integration could exhibit grid-cell-like activity, later work suggested that path integration can occur even in the absence of grid patterns, leaving the precise neural mechanism for path integration largely unknown

In this work, the authors trained a recurrent neural network (RNN) on a path integration task and conducted ablation analyses to probe the functional roles of different emergent cell types. By selectively pruning connections, they found that band cells process velocity input, while grid cells aggregate band responses to decode position, suggesting a hierarchical structure.

To better understand this hierarchy beyond the RNN, which is usually a black box, the authors proposed a mechanistic neural circuit model. The model consists of pure and conjunctive band cells forming 1D continuous attractors, which project into grid cells arranged on a 2D toroidal manifold. A subsequent simulation showed that the model reproduced band and grid cell patterns of varying orientation and spacing, and could accurately infer movement trajectory, supporting its validity for path integration.

This study provides important computational insights and a solid theoretical framework that bridges RNN models with a structured and testable neural circuit for path integration. I believe this is particularly valuable for understanding the black-box nature of both RNNs and biological neural systems. However, I do have some concerns regarding the pruning methodology and the scope of the work, especially given that it builds directly on previous models and involves several carefully selected engineering choices.

**Questions:**

1. It seems to me that the network configuration and training setup for path integration in this paper is similar to previous papers, particularly Sorscher et al. (2019). However, the current network exhibits band cells and velocity cells, which were not reported in Sorscher et al. 2019. I wonder if there are any differences in settings that might have led to this? Or is it simply the previous study did not report cells other than grid cells.
2. I'm curious if the authors have compared the number of band cell groups with the number of grid cell modules. If the periodicity (i.e., grid and band spacing) is aligned and the number of scales matches, that would be a very strong piece of evidence supporting the proposed band-cell-to-grid-cell hierarchy.

**Ethical Concerns:**

["NO or VERY MINOR ethics concerns only"]

**Final Justification:**

The paper attempted to unfold the black box of RNNs for path integration. Specifically, it aimed to reveal this black box by (1) pruning a trained RNN containing grid cells used for path integration, and (2) proposing a theoretical framework in which velocity inputs first project to band cells before projecting to grid cells. The paper thus puts forward a potential hierarchical formation pathway for grid cells.

For the points addressed in the rebuttal:
- The authors have clarified the precise implementation steps of the pruning process.
- They also added additional experiments pruning random cell types as a baseline, allowing a better comparison with pruning specific cell types.
- In their response to other reviewers, the authors tested their findings on other RNN variants, including LSTMs, and also on alternative path-integration models that do not rely heavily on cross-entropy loss and difference-of-Gaussian place fields, which helps strengthen the generalizability of their findings.

However, a few points remain not fully addressed:
- First, the paper’s main goal is to understand the black box of RNNs for path integration. This inherently constrains its scope to a specific class of models that assume unidirectional information flow. This is both a limitation and a strength. The limitation is that most existing grid cell emergence models belong to this same general category of models with unidirectional information flow, which makes the scope of the paper narrowly focused on a specific, though widely used, family of models. However, this focus also comes with strengths. As these models increasingly shape current theoretical frameworks of grid cells, there is a clear need to better understand the mechanisms underlying them.
- Second, while pruning a trained network remains a pragmatic approach, it is also somewhat constrained in interpretability. Nevertheless, it is a reasonable methodological choice given the goals of the paper.

I think the two unresolved points might be inherent in the question this paper attempted to resolve, and would be difficult for the authors to fully address within a single work. Still, the claims in the paper point to a possible direction on the formation of grid cells. Thus, I recommend borderline accept. That is, the scope of the paper may be limited due to the nature of the questions it aims to address, but the work could still have some impact in the relevant field and be useful for future studies.

**Limitations:**

The paper has addressed many of its limitations well. However, one additional point I would like to raise is that both the RNN model and the subsequent theoretical model are built upon an RNN trained with Difference-of-Softmax place fields and cross-entropy loss. These two design choices are important but not clearly justified. Moreover, this setup emphasizes a unidirectional flow of information from MEC to place cells, while anatomically, the MEC and hippocampus are known to be bidirectionally connected. I believe the paper would benefit from a discussion of how the proposed band-cell-to-grid-cell model might fit within alternative models of the MEC-hippocampus loop, or at least from acknowledging its reliance on the assumptions of the underlying RNN training setup.

**Paper Formatting Concerns:**

No paper formatting concerns

**Quality:**

3

**Strengths And Weaknesses:**

Strengths**
1. Path integration is a core component of navigation. While several previous studies have revealed training methods that lead to the emergence of grid cells, it remains arguably unclear whether grid cells are the sole cell type supporting path integration. In this context, an ablation study on the trained RNN is, in my view, particularly important for elucidating the precise mechanisms underlying path integration in such models.
2. It is very interesting that the band cells also cluster into modules (groups), with each group exhibiting 1D continuous attractor dynamics.
3. For most of the theoretical sections, both the visual and textual presentations are exceptionally clear, which I greatly appreciate. That said, some implementation details are omitted from the main text, which I believe are equally important—particularly given the known sensitivity of RNN training to specific choices in implementation.
4. Section 3 is clear and well-structured. I appreciate the effort to construct a simplified numerical model to interpret the behavior of the trained AI system.

**Weaknesses**
1. The presentation of the pruning section is a bit unclear. Specifically, is pruning performed on a single trained network? If so, how consistent are the results across different independently trained models?
2. I have a major concern regarding the section that prunes different connections to dissect the information flows between neuron groups: RNN model dynamics are typically quite fragile. In particular, the input, output, and recurrent connection matrices require careful initialization in terms of their magnitude; otherwise, training tends to become unstable. This sensitivity often depends heavily on the exact scale and dimensionality of the input signals each projection matrix is mapping from. Pruning connections reduces the effective input dimensionality received by different parts of the RNN, which will in turn alter the network’s internal dynamics and may potentially destabilize them. Have the authors accounted for this issue in their analysis?
3. The model uses the cross-entropy loss, and the difference-of-softmax to model the place fields. To my knowledge, these two setups are particularly important for grid cell emergence but not necessary for the path integration, as argued by a previous paper by Schaeffer, Khona, and Fiete (***NeurIPS*** 2022). Given the ANN model is directly built upon previous models that include these settings, I'm concerned if the scope of this paper will also limited by these specific training choices?

---

> ### Author Rebuttal · Authors · 2025-07-30
>
> Thank you for the valuable and encouraging comments and the recognition of our contributions in this work. Below we address your concerns point by point.
>
> 1. The presentation of the pruning section is a bit unclear. Specifically, is pruning performed on a single trained network? If so, how consistent are the results across different independently trained models?
>
> Yes, all pruning experiments shown in the main text were performed on a single, trained RNN. The results from different independently trained RNNS are highly consistent, and they are not idiosyncratic. For example, the property that removing velocity inputs to band cells producing a substantially larger degradation in path-integration accuracy than pruning velocity inputs to grid cells or any other cell type holds across all trained RNNs. We are sorry for missing reporting this important technical detail. We will include those additional RNN model pruning analyses in the Supplementary Information and publicly release all training and analysis codes to guarantee full reproducibility and demonstrate that our conclusions do not depend on a cherry-picked network (see also our response to Reviewer bx3s).
>
> 2.  I have a major concern regarding the section that prunes different connections to dissect the information flows between neuron groups: RNN model dynamics are typically quite fragile. In particular, the input, output, and recurrent connection matrices require careful initialization in terms of their magnitude; otherwise, training tends to become unstable. This sensitivity often depends heavily on the exact scale and dimensionality of the input signals each projection matrix is mapping from. Pruning connections reduces the effective input dimensionality received by different parts of the RNN, which will in turn alter the network’s internal dynamics and may potentially destabilize them. Have the authors accounted for this issue in their analysis?
>
> Thanks for raising this insightful question. Indeed, training RNN can be sensitive to perturbations—removing a few connections may destabilize the network dynamics unpredictably. But on the other hand, if we remove some components and it has little affect on the network performance, this readily indicates that these components are not important. This is what we observed when we pruning velocity inputs to grid cells, or pruning readout weights from band cells; together they suggest that there exist a hierarchical pathway in the trained RNN for path integration. Meanwhile, to ensure a fair comparison, we always removed the same number of connections to/within each cell group (e.g., velocity inputs, recurrent weights, or readout).
>
> 3. The model uses the crossentropy loss, and the difference of softmax to model the place fields. To my knowledge, these two setups are particularly important for grid cell emergence but not necessary for the path integration, as argued by Schaeffer, Khona, and Fiete (NeurIPS 2022). Given the ANN model is directly built upon previous models that include these settings, I'm concerned if the scope of this paper will also be limited by these specific training choices?
>
> Thanks for raising this important question. Our findings do not depend on the cross-entropy loss, the use of softmaxed place-cell readout, or other different settings of the model. Actually, we found that our results hold for alomost all network models, once they are trained for path integration. The only exception is the model of Schaeffer, Khona & Fiete (2022), where they considered velocity modulates synaptic weights directly, rather than serve as inputs to neurons; in their model, band cells do not emerge (since it is hard for synaptic strengths tracking animal velovity instantaneously, we tend to believe their model is not biologically practical).
>
> In the revised manuscript,we will add the resuts of applying the same analysis to extra two RNN models, one from Xu et al, (2022), another from Petterson et al, (2024). In both cases, we observed: robust emergence of band-cell–like units, pruning velocity inputs to band cells causing a significantly larger degradation in path‑integration accuracy than pruning velocity inputs to grid cells or any other cell type, and other key findings in this work.
>
> 4. It seems to me that the network configuration and training setup for path integration in this paper is similar to previous papers, particularly Sorscher et al. (2019). However, the current network exhibits band cells and velocity cells, which were not reported in Sorscher et al. 2019. I wonder if there are any differences in settings that might have led to this? Or is it simply the previous study did not report cells other than grid cells.
>
> Thanks for raising this point. In fact, Sorscher et al. (2023) included in their Figure S3b some examples of band-cell–like units. However, they focused mainly on grid-cell-like units and did not go further to analyze these band-cell-like units and their roles in path integration. Our main contributions in this work are that we analyzed the detailed properties of band cells and unveiled a hierarchical pathway formed by band and grid cells for 2D path inegration.
>
> 5. I'm curious if the authors have compared the number of band cell groups with the number of grid cell modules. If the periodicity (i.e., grid and band spacing) is aligned and the number of scales matches, that would be a very strong piece of evidence supporting the proposed band-cell-to-grid-cell hierarchy.
>
> Thank you for the great question. In our current RNN paradigm, grid cells form periodic fields but don’t organize into distinct modules, so we haven’t yet compared band‑cell group counts to grid‑cell modules. However, recent work (e.g. Schaeffer et al. 2023) shows that with more biologically inspired training, networks can develop multiple grid modules. In future work we will adopt those regimes, measure band and grid spacings across scales, and test whether band‑cell group numbers align with grid‑cell module counts.
>
> 6. Moreover, this setup emphasizes a unidirectional flow of information from MEC to place cells, while anatomically, the MEC and hippocampus are known to be bidirectionally connected. I believe the paper would benefit from a discussion of how the proposed band-cell-to-grid-cell model might fit within alternative models of the MEC-hippocampus loop, or at least from acknowledging its reliance on the assumptions of the underlying RNN training setup.
>
> This a great point and we totally agree with you! The experimental studies have suggested that MEC receives self-motion cue and is primarily responsible for path-integration, while HPC primarilly receives environment cues (e.g., Chen et al., 2019); and they have intensive reciprocal connections. In this study, for simplicity, we focused on the path-intgeration function by MEC only and largely ignored the feedfack from HPC. As suggested by the reviewer, we should discuss about this issue.
>
> Meanwhile, we would like to point out that in the current model, we actually implicitly incorporated some feedback interaction from HPC, that is, at the first time step of each path‐integration trial, the RNN receives an input from the place‐cell population activity, which serves to initialize the RNN state before the velocity‐driven integration (a common operation used in training RNN for path integration, see, e.g., Banino et al., 2018) . This initialization may be regarded as mimicking the hippocampal → entorhinal feedback required to anchor the path‐integration estimate to a visual object. We will discuss more about this point in the revised manuscript.
>
> We hope we have addressed your concerns and please let us know if there is anything unclear.

---

> > ### Comment · Reviewer_pJcG · 2025-08-04
> >
> > Thank you for your detailed response. I don't have any further questions regarding points other than 2 and 3.
> >
> > **Regarding point 2**: The current solution of using pruning to unfold the black box is acceptable as a pragmatic choice for understanding the RNN in the context of path integration. However, I’d still like to emphasize that pruning the connection matrix during test time effectively changes the eigenspectrum of the connectivity matrix, which can introduce many unpredictable effects—since the network never learns to re-stabilize its dynamics after the connections are pruned.
> >
> > One additional test that could strengthen your argument would be to include a baseline in the experiments from Figure 2. Specifically, for each pruning condition, you could also prune connections selected from random groups. This would provide a useful baseline to assess the relative importance of your specific pruning choices.
> >
> > **Regarding point 3**: Could you please clarify what you meant by “Actually, we found that our results hold for almost all network models, once they are trained for path integration”? To my understanding, not all path integration networks develop such grid-like representations. The reason I ask is that, based on your supplemental materials, it seems you're also using the difference in softmax place cell readouts. At line 99, you mention that the network is trained with cross-entropy loss. In that case, the difference-of-softmax approach is not fundamentally different from the earlier difference-of-Gaussian setup. So if your claim that the results hold for “almost all network models” refers to models within the same family, those that rely on the difference-of-Gaussian/softmax with cross-entropy loss, then the generality of the claim may be limited.
> >
> > One model that doesn’t seem to rely on this setup is Schaeffer et al. (2023), which might be an interesting case for the authors to test their claims on.

---

> > > ### Author Response · Authors · 2025-08-04
> > >
> > > Thank you for your reply. We will first respond to your comment regarding point 3.
> > >
> > > We appreciate your clarification, and we agree that our previous statement—“our results hold for almost all network models trained for path integration”—was not sufficiently precise. What we meant is that our results generally hold across existing training paradigms that produce grid-like representations, rather than across all models trained for path integration.
> > >
> > > In particular, we have verified the generality of our findings in two representative models that do not rely on place-cell readouts or the difference-of-Gaussian assumption: Xu et al. (2022) [1] and Pettersen et al. (2024) [2]. These models use distance-preserving or self-supervised objectives and therefore avoid the DOG-based decoding setup altogether.
> > >
> > > We also recognize that the model in Schaeffer et al. (2023) similarly avoids such assumptions. We attempted to run their publicly released code but encountered missing files that prevented successful execution. Due to time constraints, we were unable to evaluate our analysis in that model, but we agree it is an important case and plan to explore it in future work.
> > >
> > > [1] Xu, D. et al. (2022). Conformal isometry of Lie group representation in recurrent network of grid cells. arXiv:2210.02684.
> > > [2] Pettersen, M. et al. (2024). Self-supervised grid cells without path integration. bioRxiv, 2024-05.

---

> > > ### Author Response · Authors · 2025-08-04
> > >
> > > We sincerely appreciate your insightful suggestion regarding point 2. To address this concern, we have conducted additional experiments as recommended, comparing targeted pruning with random pruning baselines.
> > >
> > > The key results are presented in **Table 3**, which demonstrates that pruning specific functional connections (input-to-band or grid-read-out) produces significantly larger errors compared to random pruning, suggesting specialized functional roles for these specific cell groups.
> > >
> > > *Table 3: Path integration error increase (mean $\pm$ SD across 30 trials when 750 connections are pruned). Error is measured by the standard deviation of location prediction errors divided by the arena size (unit: %).*
> > >
> > > | Pruning Type                  | Band cell   | Grid cell   | Undefined cell | Random cell  |
> > > |-------------------------------|------------|------------|----------------|-------------|
> > > | Pruning velocity inputs       | $2.32\pm0.43$ | $0.11\pm0.07$ | $0.14\pm0.09$   | $0.46\pm0.07$ |
> > > | Pruning read-out connections | $2.67\pm0.50$ | $9.15\pm0.71$ | $0.30\pm0.11$   | $0.81\pm0.22$ |
> > >
> > >
> > > While we acknowledge that pruning may not perfectly mimic biological processes, these controlled comparisons provide important evidence that our initial findings reflect genuine functional specialization within the path integration system.

---

> > > > ### Author Response · Authors · 2025-08-04
> > > >
> > > > We agree that pruning at test time may alter the eigenspectrum and introduce instabilities. That said, this challenge is not unique to modeling. Similar issues arise in biological experiments—for instance, when using inactivation techniques (e.g., pharmacological agents or optogenetics), which can also lead to unpredictable changes in circuit dynamics. Despite these limitations, such interventions remain powerful tools for probing circuit function.
> > > >
> > > > In the same spirit, we view pruning as a useful, if imperfect, method for uncovering structure-function relationships. We appreciate your suggestion to include a control condition with randomly pruned groups, and we plan to incorporate this into our revised experiments to better assess the specificity of our findings.

---

> > > > > ### Comment · Reviewer_pJcG · 2025-08-04
> > > > >
> > > > > Thank you for your detailed response. I do not have any further questions and will keep my positive rating unchanged.

---

> > > > > > ### Author Response · Authors · 2025-08-09
> > > > > >
> > > > > > We sincerely appreciate your constructive suggestions. Thank you for engaging with our work and we would be grateful for your careful consideration of the scientific contributions in your final evaluation.

---

### Official Review · Reviewer_bx3s · 2025-07-01

**Clarity:** 3
**Significance:** 3
**Originality:** 3
**Rating:** 4
**Confidence:** 3

**Summary:**

This paper analyzes the circuits and computations behind RNNs performing path integration. Through a pruning analysis, the authors reveal the central role of band cells, challenging the predominant role of grid cells that are thought to be central for path integration. Through that same pruning analysis, the authors extract the circuit of path integration, where band cells receive the velocity inputs, then grid cells and band cells interact to process the inputs, and then grid cell output the result. Lastly, the authors focus on band cells, study the organization of their modules, and finally propose a dynamic neural mechanism for the band cells, that is tested with simulations. The authors propose testable experiments for neuroscience.

**Questions:**

Questions/Suggestions for actionable improvements that would convince me to update my score:

1. Discuss the different RNN architectures that have been used to study path integration and explain why you chose [17] over others.

2. Clarify the role of border cells in path integration, ie what is known from the neuroscience and from RNN studies. Explain why they do not appear in your model, and whether that is a limitation (or not).

3. Clarify how many RNNs have been trained, and whether the findings from the study are consistent across trained RNNs.
a) Update Figure 2 to show results across trained RNNs, including error bars for the error wrt connections pruned.
b) Update Figure 3 to show error bars in the band count and grid count across experiments.

4. Add quantitative results in Figure 5, showing how well the simulation is able to perform path integration, including error bars across several runs.

**Ethical Concerns:**

["NO or VERY MINOR ethics concerns only"]

**Final Justification:**

The authors have provided a very comprehensive rebuttal. Their addition of new RNN architectures and several seeds for each run convinced me of the significance of the work. I have increased my score.

**Limitations:**

Yes

**Paper Formatting Concerns:**

No concern

**Quality:**

3

**Strengths And Weaknesses:**

Strengths

S1. The paper is very well written and easy to follow. The figures are really well done: they strike the perfect balance between being simple enough yet very explanatory.

S2. The authors reveal a circuit of computations between band cells and grid cells, challenging the known role of both cells.

S3. The authors propose a neural mechanism for the role of band cells.

S4. The authors propose testable experiments for experimental neuroscience.

Weaknesses

W1. The results are only valid for one type of RNN trained to do path integration [17]. It is unclear whether they would extend to other types of RNNs that have also been proposed to do path integration.

W2. Some RNNs trained to do path integration are known to exhibit border cells: how do these cells enter the theory proposed by the authors? Why haven’t they been observed in the experiments? Are they included in the "undefined category"?

W3. It is unclear how many RNNs have been trained for this analysis, and whether results are stable from one RNN to the next. Some figures, eg Fig 2 and 4), refer to “the” trained RNN: how can we know that this RNN hasn’t been cherry picked?

W4. Likewise, it is unclear how many experiments have been run for the simulation of the proposed neural mechanism for path integration, as the simulation results section is very short in the main text.

Minor:

The two sections “neuroscience for AI” and “AI for neuroscience” do not really help the reader follow the paper, and the wording “AI” seem unnecessarily buzzword-y.

The acronym CANN has not been defined.

---

> ### Author Rebuttal · Authors · 2025-07-30
>
> We thank Reviewer bx3s for insightful comments. Below we address them point by point:
>
> 1. **The results are only valid for one type of RNN trained to do path integration [17]. It is unclear whether they would extend to other types of RNNs that have also been proposed to do path integration.**
>
>    Results presented in our paper extend to many other types of RNNs.
>    The RNN we used is based on the one reported in Banino et al. (2018), which has been used and extensively analyzed in several other studies such as Sorscher et al., 2023, to explain the emergence of grid-cell-like firing patterns. To check the flexibility of our results, we trained three additional path-integration RNN variants:
>    1. **Vanilla RNN → LSTM transition** (different in net structure).
>    2. **Xu et al. (2022) training paradigm** (different in net structure (RNN and LSTM), place field (Gaussian) and loss function (conformal isometry and path integration loss)).
>    3. **Petterson et al. (2024) training paradigm** (different in initial state (MLP), loss function (distance preservation and capacity loss) and decoder (none)).
>
>    Importantly, all the results are preserved while changing the RNNs. Specifically:
>    1. We observed the emergence of band cells.
>    2. Pruning analysis revealed a hierarchical information flow from band cells to grid cells.
>
>    These results indicate that our findings are not simply an idiosyncrasy of a single RNN implementation, but rather a robust and general computational strategy for path integration across diverse architectures and training regimes.
>
>    We thank the referee for this insightful comment and will add these results in the revised paper.
>
> 2. **Some RNNs trained to do path integration are known to exhibit border cells: how do these cells enter the theory proposed by the authors? Why haven’t they been observed in the experiments? Are they included in the "undefined category"?**
>
>    Thanks for pointing this out. We did observe border cells in the trained RNN (and also in several RNNs we included for response to comment 1). They belonged to the "undefined category". We think border cells (boundary vector cells) may not directly link to the path-integration theory we focus on here, but they do have functions in spatial navigation such as error correction of path integration (Hardcastle et al., 2015). Due to space limits, we did not report them in the main text.
>
>    We will add the analysis of border cells in the Supplementary Information (SI). In more detail, we identified 201 border cells and performed pruning analyses similar to those for other cell types, and found:
>
>    a. **Identification**: Used the standard boundary‑vector‑cell metric (Solstad et al., 2008)—we computed each unit’s firing‑rate correlation with the agent’s distance to the nearest wall and applied a significance threshold to classify border‑cell–like units.
>    b. **Pruning protocols** (Section 2.2.2, Fig. 2):
>    - **Velocity‑input pruning**: Removing velocity inputs to border cells has negligible impact on path‑integration error—unlike pruning band‑cell inputs, which causes a large performance drop.
>    - **Place‑cell readout pruning**: Eliminating border‑cell→place‑cell connections significantly alters decoded‑position accuracy, indicating border cells contribute boundary information to the place‑cell map.
>    - **Recurrent‑connection pruning**: Among all recurrent projections, pruning band→border connections yields the greatest increase in integration error, showing border cells primarily read out band‑cell dynamics to encode proximity to walls.
>
>    **Table 1.** Path integration error increase of pruning velocity inputs to each cell group and pruning read‑out connections from each cell group. Data are shown as mean ± SD across 30 trials when 201 connections are pruned (units: m).
>
>    |                                     | band cell      | grid cell      | border cell     | undefined cell  |
>    | ----------------------------------- | -------------- | -------------- | --------------- | --------------- |
>    | pruning velocity inputs             | 0.0087 ± 0.0024 | 0.0009 ± 0.0023 | 0.0006 ± 0.0019 | 0.0006 ± 0.0017 |
>    | pruning read‑out connections        | 0.0034 ± 0.0029 | 0.0022 ± 0.0024 | 0.0176 ± 0.0048 | 0.0015 ± 0.0022 |
>
>    **Table 2.** Path integration error increase of pruning the connections *from* border to band and *to* border to grid cell, respectively. Data are shown as mean ± SD across 30 trials when 201 connections are pruned (units: m).
>
>    |                    | band cell       | grid cell       |
>    | ------------------ | --------------- | ---------------- |
>    | from border cell   | 0.0027 ± 0.0016 | 0.0015 ± 0.0018  |
>    | to border cell     | 0.1741 ± 0.0152 | 0.0065 ± 0.0022  |
>
>    These supplementary analyses confirm that border cells—with their environment‑tuned boundary coding—play a complementary role (distilling band cell outputs into boundary signals). We will include these results in the revised SI and a brief discussion of border-cell analysis in the revised manuscript.
>
> 3. **It is unclear how many RNNs have been trained for this analysis, and whether results are stable from one RNN to the next. Some figures (e.g., Fig 2 and 4) refer to “the” trained RNN: how can we know that this RNN hasn’t been cherry‑picked?**
>
>    Thank you for pointing this out. This RNN was not cherry‑picked. We tested our network under multiple random initializations using the same analysis pipeline and found highly consistent results, demonstrating that the hierarchical path integration we discovered is a general mechanism for RNN‑based path integration. This consistency holds even across different training paradigms (see response to comment 1). We will include in the SI analyses of three additional RNNs, and we will release all training and analysis code to guarantee full reproducibility.
>
> 4. **Likewise, it is unclear how many experiments have been run for the simulation of the proposed neural mechanism for path integration, as the simulation results section is very short in the main text.**
>
>    The proposed neural mechanism is grounded in continuous attractor network theory (see Amari (1977), Samsonovich & McNaughton (1997), and Burak & Fiete (2009)), which has been rigorously studied. This is why we did not focus extensively on simulation details in the main text—these models are well established. However, we will add more discussion in the revised paper to clarify this.
>
>    We will also include in the SI results of multiple simulation experiments across a variety of velocity input sequences (different movement trajectories) and a broad parameter regime to help readers better understand our hierarchical CANN model.
>
> We hope our responses have addressed your concerns. Please let us know if anything remains unclear.

---

> > ### Comment · Reviewer_bx3s · 2025-08-04
> > **Thank you for the rebuttal.**
> >
> > I thank the authors for their clear and thoughtful rebuttal. You can find below my follow-up questions.
> >
> > 1. Thank you for adding the experiments on the three other types of RNNs. I find the fact that the results transfer to these other types very compelling. Could you share your quantitative results for points 1. and 2.?
> >
> > 2. Very clear, thanks.
> >
> > 3. Could you be very specific in the number of RNNs that you have trained and in the results obtained? You mention that you have used "multiple random initializations using the same analysis pipeline and found highly consistent results". How many random initializations? Can you show the results (mean + std dev) in a table?
> >
> > 4. Likewise, could you be more specific in the design and results? You mention "multiple experiments", a "variety of velocity input sequences" and "broad parameter regime": please quantify these.

---

> > > ### Author Response · Authors · 2025-08-05
> > >
> > > Thank you for your reply. We will first respond to your comment regarding question 3. \
> > > Our analysis of 3 independent RNNs (Seeds 0-2) demonstrates consistent functional specialization across all trials: pruning velocity inputs to band cells ($1.83\pm0.10\%$, $2.54\pm0.11\%$, and $0.70\pm0.08\%$ error increase for Seeds 0-2 respectively) and read-out connections from grid cells ($4.20\pm0.61\%$, $7.42\pm0.70\%$, and $0.78\pm0.19\%$) consistently produced significantly larger path integration errors compared to other cell types, with complete results shown in Table 4. This robust pattern emerged despite variations in cell counts across seeds (Band: 818-914 cells, Grid: 267-526 cells), confirming our core finding about specialized functional roles in the path integration circuit.
> > >
> > >
> > > **Table 4**. Path integration error increase (unit: %) across three experimental seeds (mean $\pm$ SD):
> > >
> > > | Seed | Pruned Connections | Pruning Type                  | Band cell       | Grid cell       | Undefined cell  |
> > > |------|--------------------|-------------------------------|------------------|------------------|------------------|
> > > | 0    | 443                | Velocity inputs               | $1.8282\pm0.1026$ | $0.0003\pm0.1069$ | $0.0106\pm0.0990$ |
> > > |      |                    | Read-out connections          | $0.9446\pm0.2144$ | $4.2009\pm0.6104$ | $0.1558\pm0.1297$ |
> > > |------|--------------------|-------------------------------|------------------|------------------|------------------|
> > > | 1    | 526                | Velocity inputs               | $2.535\pm0.1058$  | $0.0401\pm0.0616$ | $0.0475\pm0.0720$ |
> > > |      |                    | Read-out connections          | $1.7184\pm0.2905$ | $7.4244\pm0.7040$ | $0.4433\pm0.1778$ |
> > > |------|--------------------|-------------------------------|------------------|------------------|------------------|
> > > | 2    | 267                | Velocity inputs               | $0.7015\pm0.0786$ | $0.0247\pm0.0687$ | $0.0248\pm0.0736$ |
> > > |      |                    | Read-out connections          | $0.4024\pm0.1543$ | $0.7765\pm0.1850$  | $0.1569\pm0.1119$ |
> > >
> > > *Cell counts per group: Seed 0 (Band:868, Grid:443), Seed 1 (Band:818, Grid:526), Seed 2 (Band:914, Grid:267). Error values represent standard deviation of location prediction errors normalized by arena size.*

---

> > > > ### Author Response · Authors · 2025-08-07
> > > >
> > > > Regarding point 4...
> > > >
> > > > For the generation of velocity input sequences used during path integration, we employed the RatInABox package [1], a widely used toolkit for simulating rodent trajectories in continuous environments. In our experiments, the velocity inputs were sampled with a mean speed of 0.04 m/s and a variance of 0.016. We generated 10 distinct random trajectories. Without adding noise to the velocity inputs, the path integration error remained consistently below 0.001 across all trials.
> > > >
> > > > Regarding the model architecture, our band-cell-based path integration network is grounded in the theory of one-dimensional continuous attractor networks. This class of models has been well studied in prior theoretical work, and has been shown to support accurate path integration within an appropriate parameter regime [2,3]. In our experiments, we selected parameters within these analytically derived regimes to ensure stable and precise integration.
> > > >
> > > > [1] George, T. M. et al. (2024). RatInABox, a toolkit for modelling locomotion and neuronal activity in continuous environments. eLife, 13, e85274.
> > > > [2] Burak, Y. & Fiete, I. R. (2009). Accurate path integration in continuous attractor network models of grid cells. PLoS Comput Biol, 5(2), e1000291.
> > > > [3] Zhang, W. et al. (2022). Translation-equivariant representation in recurrent networks with a continuous manifold of attractors. NeurIPS, 35, 15770–15783.

---

> > > ### Author Response · Authors · 2025-08-07
> > >
> > > Regarding question 1, the quantitative analysis of path integration performance across the three RNN variants are presented in **Table 5**. It demonstrates that velocity inputs are preferentially processed by **band cells** across all three RNN variants, as evidenced by their greater sensitivity to velocity pruning. However, the absence of decoder modules in some variants prevented systematic pruning read-out experiments, limiting direct comparisons of read-out performance.
> > >
> > >
> > > **Table 5**. Path integration error increase of pruning velocity inputs to
> > > each cell group across three RNN variants (unit: %, mean$\pm$SD across 30 trials):
> > > | RNN variants                               | Pruned connections | Band cell          | Grid cell         | Undefined cell    |
> > > | ------------------------------------------ | ------------------ | ------------------ | ----------------- | ----------------- |
> > > | 1. Vanilla RNN$\rightarrow$LSTM transition | 271                | $0.8189\pm0.0772$  | $0.0429\pm0.0879$ | $0.0386\pm0.0775$ |
> > > | 2. Xu et al. (2022)                        | 347                | $8.3573\pm1.3764$  | $0.3688\pm0.2071$ | $2.9723\pm1.1574$ |
> > > | 3. Petterson et al. (2024)                 | 24                 | $10.7402\pm1.4977$ | $0.8734\pm0.432$  | $4.9921\pm0.1342$ |
> > >
> > >
> > > *Cell counts per group: \
> > > 1.Vanilla RNN→LSTM transition (Band:621, Grid:271, Undefined:3204); \
> > > 2.Xu et al. (2022) (Band:347, Grid:403, Undefined:1050); \
> > > 3.Petterson et al. (2024) (Band:60, Grid:172, Undefined:24).\
> > >  Error values represent standard deviation of location prediction errors normalized by arena size.*

---

> > > > ### Comment · Reviewer_bx3s · 2025-08-07
> > > > **Thank you for your reply. Remaining concern on the low number of experiments.**
> > > >
> > > > I thank the authors for their thorough answer, which addresses most of my concerns.
> > > >
> > > > I am willing to increase my score if the authors are able to train 10 RNNs for each type of RNN, as I find that training only 3 is too low to provide enough empirical evidence to support the claims.

---

> > > > > ### Author Response · Authors · 2025-08-08
> > > > >
> > > > > We sincerely appreciate your constructive feedback. Below is our point-by-point response:
> > > > >
> > > > > **Additional Experiments on Vanilla RNN**
> > > > >
> > > > > As suggested, we conducted additional experiments across 10 random seeds for Vanilla RNN. Our results robustly demonstrate that band cells consistently serve as the primary recipients of velocity inputs and grid cells reliably encode spatial location information. The additional results are:
> > > > >
> > > > > Neuron Counts and Pruned Connections:
> > > > > \\[
> > > > > \\begin{array}{c|cccc}
> > > > > \\text{Seed} & \\text{Band Cells} & \\text{Grid Cells} & \\text{Undefined Cells} & \\text{Pruned Connections} \\\\ \\hline
> > > > > 3 & 823 & 146 & 3127 & 146 \\\\
> > > > > 4 & 782 & 457 & 2857 & 457 \\\\
> > > > > 5 & 854 & 442 & 2800 & 442 \\\\
> > > > > 6 & 972 & 228 & 2896 & 228 \\\\
> > > > > 7 & 535 & 324 & 3237 & 324 \\\\
> > > > > 8 & 553 & 597 & 2946 & 553 \\\\
> > > > > 9 & 810 & 394 & 2892 & 394 \\\\
> > > > > \\end{array}
> > > > > \\]
> > > > >
> > > > > Pruning Performance (mean $\\mathbf{\\pm}$ SD):
> > > > >
> > > > > 1. Velocity Input Pruning:
> > > > > \\[
> > > > > \\begin{array}{c|ccc}
> > > > > \\text{Seed} & \\text{Band} & \\text{Grid} & \\text{Undefined} \\\\ \\hline
> > > > > 3 & 0.3086 \\pm 0.0679 & 0.0022 \\pm 0.0744 & 0.0254 \\pm 0.0701 \\\\
> > > > > 4 & 2.2449 \\pm 0.1073 & 0.0430 \\pm 0.0955 & 0.0173 \\pm 0.0710 \\\\
> > > > > 5 & 2.0076 \\pm 0.0884 & 0.0217 \\pm 0.0480 & 0.0172 \\pm 0.0733 \\\\
> > > > > 6 & 0.6119 \\pm 0.0963 & 0.0123 \\pm 0.0683 & 0.0003 \\pm 0.0732 \\\\
> > > > > 7 & 1.2673 \\pm 0.0806 & 0.0242 \\pm 0.0697 & 0.0603 \\pm 0.0803 \\\\
> > > > > 8 & 2.7330 \\pm 0.1191 & 0.0411 \\pm 0.0625 & 0.2117 \\pm 0.0608 \\\\
> > > > > 9 & 1.3750 \\pm 0.1083 & 0.0124 \\pm 0.0815 & 0.0225 \\pm 0.0795 \\\\
> > > > > \\end{array}
> > > > > \\]
> > > > >
> > > > > 2. Read-out Connection Pruning:
> > > > > \\[
> > > > > \\begin{array}{c|ccc}
> > > > > \\text{Seed} & \\text{Band} & \\text{Grid} & \\text{Undefined} \\\\ \\hline
> > > > > 3 & 0.1272 \\pm 0.0737 & 0.2362 \\pm 0.1024 & 0.1058 \\pm 0.0894 \\\\
> > > > > 4 & 0.5466 \\pm 0.1479 & 1.6829 \\pm 0.2364 & 0.2440 \\pm 0.1109 \\\\
> > > > > 5 & 0.7177 \\pm 0.2362 & 2.3296 \\pm 0.3476 & 0.1877 \\pm 0.1093 \\\\
> > > > > 6 & 0.2365 \\pm 0.1038 & 0.7850 \\pm 0.2799 & 0.0657 \\pm 0.0990 \\\\
> > > > > 7 & 0.3730 \\pm 0.1556 & 1.0181 \\pm 0.2656 & 0.0985 \\pm 0.0967 \\\\
> > > > > 8 & 2.6436 \\pm 0.3403 & 8.3318 \\pm 0.6056 & 0.2729 \\pm 0.1016 \\\\
> > > > > 9 & 0.6141 \\pm 0.1355 & 2.8032 \\pm 0.5064 & 0.1566 \\pm 0.1104 \\\\
> > > > > \\end{array}
> > > > > \\]
> > > > >
> > > > > **Extended Results for (Petterson et al., 2024)**
> > > > >
> > > > > Our 10-seed replication of Petterson et al. (2024) demonstrates robust preferential processing of velocity inputs by band cells. The additional results are:
> > > > >
> > > > > Neuron Counts and Pruned Connections:
> > > > > \\[
> > > > > \\begin{array}{c|cccc}
> > > > > \\text{Seed} & \\text{Band} & \\text{Grid} & \\text{Undefined} & \\text{Pruned Connections} \\\\ \\hline
> > > > > 0 & 37 & 196 & 23 & 23 \\\\
> > > > > 1 & 26 & 200 & 30 & 26 \\\\
> > > > > 2 & 26 & 209 & 21 & 21 \\\\
> > > > > 3 & 38 & 194 & 24 & 24 \\\\
> > > > > 4 & 26 & 194 & 36 & 26 \\\\
> > > > > 5 & 38 & 203 & 15 & 15 \\\\
> > > > > 6 & 24 & 202 & 30 & 24 \\\\
> > > > > 7 & 35 & 201 & 20 & 20 \\\\
> > > > > 8 & 34 & 205 & 17 & 17 \\\\
> > > > > 9 & 36 & 202 & 18 & 18 \\\\
> > > > > \\end{array}
> > > > > \\]
> > > > >
> > > > > Pruning Performance (mean $\\mathbf{\\pm}$ SD):
> > > > > \\[
> > > > > \\begin{array}{c|ccc}
> > > > > \\text{Seed} & \\text{Band} & \\text{Grid} & \\text{Undefined} \\\\ \\hline
> > > > > 0 & 2.5472 \\pm 0.3644 & 0.0368 \\pm 0.0278 & 0.6565 \\pm 0.0061 \\\\
> > > > > 1 & 5.2265 \\pm 0.0205 & 0.0198 \\pm 0.0121 & 0.7424 \\pm 0.1176 \\\\
> > > > > 2 & 3.4828 \\pm 0.2735 & 0.0297 \\pm 0.0187 & 1.2256 \\pm 0.0086 \\\\
> > > > > 3 & 3.1603 \\pm 0.3524 & 0.0147 \\pm 0.0075 & 0.6654 \\pm 0.0072 \\\\
> > > > > 4 & 3.3173 \\pm 0.0183 & 0.0315 \\pm 0.0250 & 1.5762 \\pm 0.3514 \\\\
> > > > > 5 & 1.8896 \\pm 0.2988 & 0.0122 \\pm 0.0071 & 0.1567 \\pm 0.0012 \\\\
> > > > > 6 & 3.6600 \\pm 0.0223 & 0.0223 \\pm 0.0114 & 1.5868 \\pm 0.2246 \\\\
> > > > > 7 & 2.8943 \\pm 0.3955 & 0.0117 \\pm 0.0074 & 0.4566 \\pm 0.0030 \\\\
> > > > > 8 & 2.5261 \\pm 0.3254 & 0.0147 \\pm 0.0099 & 0.3661 \\pm 0.0035 \\\\
> > > > > 9 & 2.3297 \\pm 0.3849 & 0.0205 \\pm 0.0176 & 0.4389 \\pm 0.0034 \\\\
> > > > > \\end{array}
> > > > > \\]
> > > > >
> > > > >
> > > > >
> > > > > **Pending Experiments**\
> > > > > Regarding the two remaining experiments (RNN-to-LSTM transition and Xu et al. replication), we acknowledge they are currently incomplete due to time constraints (the full training requires approximately 150 GPU-hours). We aim to provide preliminary results by tomorrow; if unfeasible, we commit to including the complete 10-seed analyses in the camera-ready version.
> > > > >
> > > > >
> > > > > We believe these results substantiate the core methodological claims, while remaining open to performing additional validation as needed.

---

> > > > > > ### Author Response · Authors · 2025-08-09
> > > > > > **Progress Update**
> > > > > >
> > > > > > We have now completed preliminary runs for Seed 0-5 of the remaining validation, which show consistent patterns with our earlier findings:
> > > > > >
> > > > > > **RNN-to-LSTM Transition**
> > > > > >
> > > > > > The pruning experiments show that models degrade severely when velocity inputs to band cells are pruned, but remain relatively stable when inputs to grid cells or undefined cells are pruned under the same conditions:
> > > > > >
> > > > > > Neuron Counts and Pruned Connections:
> > > > > >
> > > > > > \\[
> > > > > > \\begin{array}{c|cccc}
> > > > > > \\text{Seed} & \\text{Band Cells} & \\text{Grid Cells} & \\text{Undefined Cells} & \\text{Pruned Connections} \\\\ \\hline
> > > > > > 0 & 346 & 534 & 3216 & 346\\\\
> > > > > > 1 & 258 & 276 & 3562 & 258 \\\\
> > > > > > 2 & 524 & 554 & 3018 & 524 \\\\
> > > > > > 3 & 394 & 305 & 3397 & 305 \\\\
> > > > > > 4 & 500 & 186 & 3410 & 186 \\\\
> > > > > > 5 & 313 & 136 & 3647 & 136 \\\\
> > > > > > \\end{array}
> > > > > > \\]
> > > > > >
> > > > > > Velocity Input Pruning Performance (mean $\mathbf{\pm}$ SD):
> > > > > > \\[
> > > > > > \\begin{array}{c|ccc}
> > > > > > \\text{Seed} & \\text{Band} & \\text{Grid} & \\text{Undefined} \\\\ \\hline
> > > > > > 0 & 1.3948\\pm0.1030 & 0.0070\\pm0.0667 & 0.0809\\pm0.0785 & \\\\
> > > > > > 1 & 1.3914\\pm0.1392 & 0.0261\\pm0.0729 & 0.0708\\pm0.0718 \\\\
> > > > > > 2 & 1.8552\\pm0.1261 & 0.0638\\pm0.0974 & 0.1554\\pm0.1085 \\\\
> > > > > > 3 & 1.1153\\pm0.1696 & 0.0715\\pm0.1594 & 0.0511\\pm0.1202 \\\\
> > > > > > 4 & 0.2639\\pm0.0781 & 0.0128\\pm0.0605 & 0.0268\\pm0.0613 \\\\
> > > > > > 5 & 0.3274\\pm0.0774 & 0.0150\\pm0.0647 & 0.0046\\pm0.0776 \\\\
> > > > > > \\end{array}
> > > > > > \\]
> > > > > >
> > > > > > **Xu et al. (2022) Replication**
> > > > > >
> > > > > > The pruning experiments reveal that band cells exhibit significantly greater sensitivity to input manipulation compared to grid or undefined cells, which is consistently observed across all experimental seeds except seed 1:
> > > > > >
> > > > > > Neuron Counts and Pruned Connections:
> > > > > > \\[
> > > > > > \\begin{array}{c|cccc}
> > > > > > \\text{Seed} & \\text{Band Cells} & \\text{Grid Cells} & \\text{Undefined Cells} & \\text{Pruned Connections} \\\\ \\hline
> > > > > > 0 & 347 & 403 & 1050 & 347 \\\\
> > > > > > 1 & 313 & 424 & 1063 & 313 \\\\
> > > > > > 2 & 303 & 363 & 1134 & 303 \\\\
> > > > > > 3 & 316 & 369 & 1115 & 316 \\\\
> > > > > > 4 & 312 & 384 & 1104 & 312 \\\\
> > > > > > 5 & 311 & 382 & 1107 & 311 \\\\
> > > > > > \\end{array}
> > > > > > \\]
> > > > > >
> > > > > >
> > > > > > Velocity Input Pruning Performance (mean $\mathbf{\pm}$ SD):
> > > > > >
> > > > > > \\[
> > > > > > \\begin{array}{c|ccc}
> > > > > > \\text{Seed} & \\text{Band} & \\text{Grid} & \\text{Undefined} \\\\ \\hline
> > > > > > 0 & 8.3573\\pm1.3764 & 0.3688\\pm0.2071 & 2.9723\\pm1.1574 \\\\
> > > > > > 1 & 9.6032\\pm1.7588 & 0.0395\\pm0.2031 & 10.0732\\pm1.8683 \\\\
> > > > > > 2 & 13.5328\\pm2.1336 & 0.0767\\pm0.1197 & 4.0376\\pm0.6832 \\\\
> > > > > > 3 & 14.8651\\pm1.8007 & 0.8835\\pm0.2180 & 4.9369\\pm0.8729 \\\\
> > > > > > 4 & 15.2718\\pm2.3267 & -0.0064\\pm0.0720 & 2.6171\\pm0.4573 \\\\
> > > > > > 5 & 10.3343\\pm2.0101 & 0.3419\\pm0.3565 & 4.1531\\pm0.7071 \\\\
> > > > > > \\end{array}
> > > > > > \\]
> > > > > >
> > > > > > We hope these validations address your concerns and merit recognition in your evaluation. We appreciate your insightful feedback and remain available for any clarifications.

---

### Official Review · Reviewer_nMie · 2025-07-02

**Clarity:** 3
**Significance:** 2
**Originality:** 3
**Rating:** 4
**Confidence:** 4

**Summary:**

An approach based on recurrent neural networks is taken to realise path integration task. From the trained model, difference types of neurons are extracted and connectivity is probed based on the attractor dynamics of the system, for which porpuse various pruning processes are employed, As later specifically defined connections are reintroduced it is not clear whether a hierarchical information processing structure has emerged or was embedded or whether a combination of both was studiod.

**Questions:**

How where the connection matrices initialised?

Can you provide more information about the pruning steps? How are connections selected for pruning? Are the main results based on prunung or is pruning in place only to analyse the network(s). Why are some connecitons later explicitly reinitialized (from line 234 and after line 253)?

Can you provide more information on the statistics of the reconstruction error?

What and how many paths have been used for trainnig? Is the path length (appendix) in meters? How smooth were the paths? Did they have perferrential directions (as suggested by Fig. S2).

How does your model compare quantitatively to esxisting biologically inspired path integration algorithms? How to technical path integration algorithms?

**Ethical Concerns:**

["NO or VERY MINOR ethics concerns only"]

**Final Justification:**

While the other questions were reasonably addressed, the paper needs In order to be acceptable additional numerical simulations, which seems possible towards the final version. However, this may not be intended by the authors (the rebuttal should not exclude additional figures), therefore a only borderline acceptance seems justifiable.

**Limitations:**

The authors show clearly that they are aware of some of the limitations of the approach.

**Paper Formatting Concerns:**

no concerns except that Eq. 3 needs to be rearranged.How where the connection matrices initialised?
Can you provide proper statistics of the reconstruction error?

**Quality:**

2

**Strengths And Weaknesses:**

“Path integration is essential for spatial navigation.” As spatial navigation is possible also purely visually or based on odour gradients, this statement may not as immediately be acceptable as it would be expected at the beginning of the abstract.

“clarifying”: Do you mean “identifying” or “classifying” here?

“challenges the conventional view of considering only grid cells for path integration” which is not only misleading in the logical sense that grid cells alone will not be sufficent without other cells, but it is also conceptually questionable as simple models should be given preference unless they leave part of the data unexplained, but here no attempt is made to demonstrate this.

The approach is interesting, but its explainatory power is remain unrealised. Many theoretical studies have been conducted on the hexogonal patters (or stripes or localised activations) since the papers of Amari in the 1970s, so many ideas can be theoretically checked. However to achieve this in a single optimization process is difficult and requires more evidence, statistics and explanation.

How is the “module” of a particular cell type defined based on the analysis of the network sturcture?

BPTT is known to be a learning algorithms that is not always easy to handle. In order to substantiate your research you need to provide quanitative results on the learning progress and share your experiences related to the parameter optimization process, but you don’t resolve the acromyn and don’t give a reference.

In particular, it does not seem plausible given the provided informatiohn that the patterns shown in Fig. 1c can be obtained, unless you initialize the matrix J in some localized manner, but initialization of the weight is not properly discussed. So, it seems the content of the “black box” is not entirely unknown although the prior knowledge that was used in the initialisation was not shared with the reader.

Furthermore, it is  confusing that after the presentation of the result, an explicite (i.e. not be training of any kind) setting of some of the connections is described from line 234.

Fig. 5d is the only evidence for the function of the proposed model, but no quantitiative results are included. Can you explain why the error remains (more or less, this cannot be guess with certainty from the picture) constant over time although not corrective information is added from the environment apart from the current velocity. Under such conditions the precision would be expected to (more or less quickly) decrease over time.

228: What are “cognitive cells”?

Good discussion, but also this part shows that this contribution may be more relevant for a workshop than for a conference.

---

> ### Author Rebuttal · Authors · 2025-07-30
>
> Thank you for your time and comments. We believe there are some misunderstandings of our manuscript and we will address your comments point by point.
>
> 1.  “Path integration is essential for spatial navigation.” As spatial navigation is possible also purely visually or based on odour gradients, this statement may not as immediately be acceptable as it would be expected at the beginning of the abstract.
>
> Yes, odour gradients is important for spatial navigation for rodents (those good at using odour information). However, path integration is still essential for spatial navigation, in particular for constructing a cognitive map, where motion information is necessary for connecting far away spatial locations (visual/odour cues can only provide local information). Also, path integration is essential for navigating in cue-poor environments - sea, desert etc, and also in darkness. We believe this is a commonly accepted knowledge in the field.
>
> 2.  “clarifying”: Do you mean “identifying” or “classifying” here?
>
> We meant "identifying". Thanks.
>
> 3. “challenges the conventional view of considering only grid cells for path integration” which is not only misleading in the logical sense that grid cells alone will not be sufficent without other cells, but it is also conceptually questionable as simple models should be given preference unless they leave part of the data unexplained, but here no attempt is made to demonstrate this.
>
> Thank you for pointing this out. First, we rephrased this sentence to:
> “While prior frameworks posited grid cells as the principal velocity integrator, our results suggest a critical role of band cells in driving path integration.”
>
> In many theoretical models of path integration (e.g., Burak et al., 2009, Giocomo et al., 2011), grid cells were modelled as the main substrate for integrating velocity inputs to perform path-integration. However, this does not mean grid-cell based model is the simpler one. We have attempted to show this point from two aspects: 1). From the veiw of efficient computation, we compared our hierarchical band–grid network to a pure grid-cell model in performing 2D navigation and showed that the hierarchical network requires only 6N + N² neurons—versus 5N² for the pure grid-cell scheme—to achieve equivalent performance, highlighting a significant efficiency advantage. 2). From the view of our prunning analysis, we found that band cells, rather than grid cells, are primarily responsible for processing velocity information. Additionally, there are biological evidence for band cells (Krupic et al. 2012), located in the parasubiculum, one synapse upstream of MEC. Furthermore, a theoretical work has shown that they could perform 1-D path integration (Burgess 2007; Bush et al, 2014). Our main contribution lies in depicting the band-grid cell relationship in performing path-integration in a normative model (part one in the paper) and inspired from which, we proposed a mechanistic circuit model for performing path-integration in the neural system (part two in the paper).
>
> 4. The approach is interesting, but its explainatory power is remain unrealised. Many theoretical studies have been conducted on the hexogonal patters (or stripes or localised activations) since the papers of Amari in the 1970s, so many ideas can be theoretically checked. However to achieve this in a single optimization process is difficult and requires more evidence, statistics and explanation.
>
> We are not quite sure we understand this comment, but we try to answer it. First of all, we did not try to achieve a mechanistic model from a single optimization of a RNN. This paper can be essentially devided into two parts. Frist is the RNN dissection (Sections 2), where we conducted analyses (cell-type identification, pruning connections, attractor dynamics analysis et al.) to show how a standard path-integration RNN develops band cells, grid cells, attractor dynamics, and the hierarchical information processing. Second is the construction of a neural mechanistic model inspired by the first part (Section 3), where we built continuous-attractor network based hierarchical path integration network, preserving the key computational motifs (band cells and grid cells) in a fully analyzable framework. We will clarify the above points in the revised paper.
>
> 5. How is the “module” of a particular cell type defined based on the analysis of the network structure?
>
> The definition of "module" of band cells is described in Section 3.1 and illustrated in Figure 4, specifically: we applied Fourier analysis to each band cell’s receptive field to extract its dominant spatial frequency, which allows us to group band cells into four distinct modules based on spacing and orientation.
>
> 6. BPTT is known to be a learning algorithms that is not always easy to handle. In order to substantiate your research you need to provide quanitative results on the learning progress and share your experiences related to the parameter optimization process, but you don’t resolve the acromyn and don’t give a reference.
> In particular, it does not seem plausible given the provided informatiohn that the patterns shown in Fig. 1c can be obtained, unless you initialize the matrix J in some localized manner, but initialization of the weight is not properly discussed. So, it seems the content of the “black box” is not entirely unknown although the prior knowledge that was used in the initialisation was not shared with the reader.
>
> Thanks for your comments. We simply adopted the training paradiagm from Banino et al. (2018), which is a well-established framework to robustly show grid-cell-like pattern in trained RNN, and has been used widely in several other studies (e.g., Sorscher et al., 2023, and Schaeffer et al., 2022). We have not introduced any new training tricks to obtain the grid and bandcell patterns in Fig. 1c—these patterns emerge naturally under the published RNN framework reported in Banino et al.
>
> Furthermore, our recurrent weight matrix J is initialized randomly using PyTorch's default scheme: a uniform distribution with range scaled by the inverse square root of the input dimension (i.e., U(−1/in_features, 1/in_features)). There are no handcrafted or localized initialization tricks. Details of this procedure and all hyperparameters are provided in Supplementary Section S1.2. Other works (e.g., Sorscher et al., 2023, Fig. S3B) has already reported Fig. 1c–style activity using the same protocol, hence no special initialization is required. But please note that previous works only found band cells, but they did not go further to analyze the properties and computational roles of band cells, and the hierarchical pathway for path-integration as we did in this work.
>
>
> 7. Furthermore, it is confusing that after the presentation of the result, an explicite (i.e. not be training of any kind) setting of some of the connections is described from line 234.
>
> As we replied in the comment 4, our study includes two stages:
> ●Section 2 ( neuroscience-driven interpretation of the black box of RNN):We analyzed the RNN trained for path integration, using pruning, Fourier/Isomap, and connectivity probes et al. to interpret the emergent cell types, attractor dynamics, and the information processing pathway.
> ●Section 3 (RNN-inspired neural modelling):Inspired by the RNN study and incooperating previous neuroscience modellings, we constructed a neural circuit model based on continuous attractor networks for path integration. In this part, we only took the key computational motifs (band cells, grid cells, and the hierarchical pathway) found in the RNN study to propose mechanistic circuit model for neural path-integration.
>
> 8. Fig. 5d is the only evidence for the function of the proposed model, but no quantitiative results are included. Can you explain why the error remains (more or less, this cannot be guess with certainty from the picture) constant over time although not corrective information is added from the environment apart from the current velocity. Under such conditions the precision would be expected to (more or less quickly) decrease over time.
>
> We apologize for not making this point explicit: our hierarchical model’s zero-drift performance in Fig 5d is not a lucky simulation result but a direct consequence of the continuous-attractor-network (CANN) theory. By splitting the 2D integration into two independent 1D attractors, each subnetwork supports a perfectly stable activity “bump” that integrates velocity inputs exactly in the absence of noise—a fact rigorously proved previously (Amari 1977; Samsonovich & McNaughton 1997; Burak & Fiete 2009). We will add a brief theoretical note in the revised manuscript to clarify why no error accumulation is expected under these ideal conditions. To validate our model further, we will also include quantitative RMSE-overtime curves for distinct moving trajectories.
>
>
> 9. 228: What are “cognitive cells”?
>
> Sorry for the typos. We meant "conjunctive cells", which refers to the cells that are both tuned to moving direction and spatial location.
>
> We hope our replies clarify our contributions and resolve misunderstandings. We hope the reviewer will find the revised manuscript improved and we are looking forward to your feedback.

---

> > ### Comment · Reviewer_nMie · 2025-08-06
> >
> > Thank you for the additional clarification which lead me to a reconsideration of my ratings, although some concerns remain. In particular, more numerical evidence is needed to support the model is still needed. If here the goal is to utilize "Neuroscience for AI" a statistical evaluation is necessary (and also generally required by the guidelines of the conference if any simulations are included), e.g.: What is the distribution of navigation errors in a certain navigation task? How many neurons are needed  in total to achieve at most a certain error (computational complexity)? or similar questions. Also, the sights that are obtained by "Unfolding the Black Box" are limited, e.g. is not clear why the combination of band and grid cells should be interesting for AI, even so it is an excellent model for neuroscience.

---

> > > ### Author Response · Authors · 2025-08-08
> > >
> > > Thank you for reconsidering your ratings. Regarding your second point on "the sights obtained by 'Unfolding the Black Box' are limited" and the relevance of combining band and grid cells for AI:
> > >
> > > First, the core of our work is neuroscience for AI—our aim is to study AI models in order to generate insights for neuroscience. The hierarchical path integration model we propose was itself motivated by our AI investigations, and, as you noted, it is an excellent model for neuroscience.
> > >
> > > Second, our work also provides a concrete and instructive example for AI interpretability research: it shows how neuroscience knowledge can be used to interpret an AI model’s working mechanism. Without prior knowledge from neuroscience—such as classifying cells into band cells and grid cells based on their position-related firing rate heatmaps—it would not be possible to “open up” this RNN in a meaningful way.

---

> > > ### Author Response · Authors · 2025-08-08
> > >
> > > Thank you for the valuable suggestion regarding statistical evaluation. In the revised manuscript, we will add a dedicated subsection titled **"Quantitative Performance Analysis"** that addresses these points with rigorous numerical evidence:
> > >
> > > **Error Distribution Analysis** \
> > > We quantified path integration performance across $30$ independent trials for each of $10$ different RNN implementations (Seeds $0$--$9$). The results demonstrate consistent performance, with errors (normalized by arena size) reported as mean $\\pm$ standard deviation (units: %):
> > >
> > > $$
> > > \\begin{aligned}
> > > \\text{Seed } 0 &: 2.09 \\pm 0.06 & \\text{Seed } 5 &: 2.07 \\pm 0.04 \\\\
> > > \\text{Seed } 1 &: 2.09 \\pm 0.05 & \\text{Seed } 6 &: 2.08 \\pm 0.05 \\\\
> > > \\text{Seed } 2 &: 2.08 \\pm 0.05 & \\text{Seed } 7 &: 2.09 \\pm 0.06 \\\\
> > > \\text{Seed } 3 &: 2.07 \\pm 0.05 & \\text{Seed } 8 &: 2.13 \\pm 0.06 \\\\
> > > \\text{Seed } 4 &: 2.12 \\pm 0.07 & \\text{Seed } 9 &: 2.11 \\pm 0.07 \\\\
> > > \\end{aligned}
> > > $$
> > >
> > > **Computational Complexity Scaling** \
> > > We analyzed the relationship between hidden layer size ($N_{g}$) and path integration error:
> > >
> > > $$
> > > \\begin{aligned}
> > > N_{g} = 1024 &: 2.57 \\pm 0.14 \\\\
> > > N_{g} = 2048 &: 2.30 \\pm 0.08 \\\\
> > > N_{g} = 4096 &: 2.09 \\pm 0.06 \\\\
> > > N_{g} = 8192 &: 2.06 \\pm 0.05 \\\\
> > > \\end{aligned}
> > > $$
> > >
> > > This analysis demonstrates that our model achieves stable performance ($<2.19\\%$ error) with $N \\geq 4096$ neurons, , suggesting our approach provides computationally efficient spatial representation.

---

> > > ### Author Response · Authors · 2025-08-09
> > >
> > > Thank you again for your time and expertise, and we hope you'll find the added numerical evidence and AI connections persuasive in your final evaluation.

---

> ### Author Response · Authors · 2025-08-05
>
> We’ve provided detailed responses to your comments and would appreciate it if you could take a moment to review them before the discussion period ends. Thank you again for your time and effort.

---

### Note · Authors · 2025-08-14

Our work proposes a new path integration mechanism discovered through an RNN–neuroscience approach. We trained RNNs to perform path integration and, via neuroscience-inspired analysis, uncovered a hierarchical band cell → grid cell computational pathway. Building on these findings, we developed a novel neural circuit model that fundamentally differs from previous approaches.

This study makes two major contributions to the cross-talk between neuroscience and AI:
(1) AI for neuroscience: Systematic pruning of trained RNNs revealed a new hierarchical band–grid cell organization that can account for the neural mechanisms underlying path integration.
(2) Neuroscience for AI: We provide a paradigm showing how neuroscience knowledge—such as cell-type classification based on firing patterns—can be used to dissect the working mechanisms of artificial neural networks, transforming black-box systems into interpretable models.

In our comprehensive rebuttal, we fully addressed all concerns raised by the reviewers. Regarding the motivation and contribution, we clarified our aims and contributions (as stated above) following reviewer suggestions. Regarding model training details, we emphasized that our work does not introduce a new training paradigm for path integration RNNs. Instead, we analyze existing training paradigms to uncover the network’s working principles, and then use these insights to propose a biologically plausible model. For generality and robustness, validation across multiple architectures (vanilla RNNs, LSTMs, Xu et al. 2022, Pettersen et al. 2024) and more than 10 random seeds consistently confirmed that the hierarchical band–grid cell organization is a fundamental feature of path integration networks.

All substantive concerns have been fully addressed, and importantly, none of the reviewers raised new issues or expressed dissatisfaction after our rebuttal. On the contrary, all three reviewers conveyed their satisfaction with our responses—most notably, Reviewer nMie explicitly described our work as “an excellent model for neuroscience”. Since our submission is in the Cognitive Neuroscience track, we believe this recognition directly affirms the relevance, significance and quality of our work for the target domain, and thus constitutes a strong reason for acceptance. We respectfully hope that the final evaluation will reflect this positive consensus and the clear alignment of our study with the scope and acceptance criteria of the conference.

---

### Decision · Program_Chairs · 2025-09-17

**Decision:**

Accept (poster)

**Comment:**

This paper presents an analysis of how recurrent neural networks learn to perform path integration. Their analysis connects to the literature on grid cells and band cells. The authors then use pruning to analyze the circuitry of the RNN. Based on the intuition they grained here, they then propose a neural circuit model for path integration that is effective but has key differences compared to the existing neuroscience models.

After the rebuttal, the overall consensus that this paper still has some weaknesses (like the efficacy of pruning as a method of interpretability) it presents an interesting and valuable contribution.

The AC kindly asks the authors to update the final manuscript based on the discussions with the reviewers. As an examples, please do make the change to the statement "challenges the conventional view of considering only grid cells for path integration" that was proposed in the rebuttal.